# Perirenal adipose tissue contains a subpopulation of cold-inducible adipocytes derived from brown-to-white conversion

**Houyu Zhang[1,2], Yan Li[1,2], Carlos F Ibáñez[1,3,4,5]\*, Meng Xie[4,6,7]\***

[1]Chinese Institute for Brain Research, Zhongguancun Life Science Park, Beijing, China; [2]Peking University Academy for Advanced Interdisciplinary Studies, Beijing, China; [3]Peking University School of Life Sciences, Peking-Tsinghua Center for Life Sciences, Beijing, China; [4]PKU-IDG/McGovern Institute for Brain Research, Beijing, China; [5]Department of Neuroscience, Karolinska Institute, Stockholm, Sweden; [6]Peking University School of Psychological and Cognitive Sciences, Beijing Key Laboratory of Behavior and Mental Health, Beijing, China; [7]Department of Biosciences and Nutrition, Karolinska Institute, Flemingsberg, Sweden

**\*For correspondence:**
carlos.ibanez@pku.edu.cn (CFI);
meng.xie@pku.edu.cn (MX)

**Competing interest:** The authors declare that no competing interests exist.

**Abstract** Perirenal adipose tissue (PRAT) is a unique visceral depot that contains a mixture of brown and white adipocytes. The origin and plasticity of such cellular heterogeneity remains unknown. Here, we combine single-nucleus RNA sequencing with genetic lineage tracing to reveal the existence of a distinct subpopulation of $Ucp1^-$&$Cidea^+$ adipocytes that arises from brown-to-white conversion during postnatal life in the periureter region of mouse PRAT. Cold exposure restores $Ucp1$ expression and a thermogenic phenotype in this subpopulation. These cells have a transcriptome that is distinct from subcutaneous beige adipocytes and may represent a unique type of cold-recruitable adipocytes. These results pave the way for studies of PRAT physiology and mechanisms controlling the plasticity of brown/white adipocyte phenotypes.

## eLife assessment

This study presents a **valuable** finding on the process of brown-to-white adipogenic transdifferentiation within the perirenal adipose depot. The evidence supporting the claims is **convincing**, although limited sequencing depth of single nuclei and lack of regulatory insights somewhat lessens the impact of these findings. The work will be of interest to adipose tissue biologists.

## Introduction

Adipose tissue is composed of adipocytes and a stromal vascular fraction (SVF) that contains highly heterogeneous populations of adipose stem and progenitor cells (ASPCs), vascular endothelial cells, immune cells, and small populations of other cell types (*Rodeheffer et al., 2008*; *Trim and Lynch, 2022*; *Wang et al., 2020*). Based on morphological and functional differences, adipocytes have been classified into three types, that is, white, brown, and beige adipocytes. Being the most abundant type of adipocyte, white adipocytes (WAs) contain a single large lipid droplet that occupies most of their intracellular space. They are present in visceral and subcutaneous depots and their primary function is to store excess energy in the form of triglycerides. WAs also secrete a number of hormones, including leptin, adiponectin, and several others, that regulate the function of other cells in adipose

tissue as well as other organs (*Scheja and Heeren, 2019*). In contrast, brown adipocytes (BAs) contain numerous small lipid droplets and a high density of mitochondria. Their primary function is the generation of heat, that is, thermogenesis, through uncoupling of the mitochondrial electron transport chain from ATP production to produce heat. This is mediated by uncoupling protein 1 (UCP1), a BA-specific protein located at the inner mitochondrial membrane (*Cohen and Kajimura, 2021*). Beige adipocytes (BeAs) are residents of subcutaneous white adipose tissue (WAT) depots that possess inducible thermogenic capacity, despite very low basal levels of UCP1 expression (*Petrovic et al., 2010*; *Wu et al., 2012*). Thanks to recent advances in single-nucleus RNA sequencing (snRNA-seq), transcriptomic profiling of adipocytes at single-cell resolution has become feasible. snRNA-seq analysis of mouse and human WAT depots has revealed adipocyte subpopulations that are related to lipogenesis, lipid-scavenging, and stress responses (*Emont et al., 2022*; *Sárvári et al., 2021*). Profiling of adipocytes from interscapular brown adipose tissue (iBAT) of mouse and human has identified a subpopulation that negatively regulates thermogenesis via paracrine secretion of acetate (*Sun et al., 2020*).

Visceral depots such as epididymal, mesenteric, and retroperitoneal adipose tissues are mainly made of WAs; whereas BAs and BeAs are often found in subcutaneous depots, such as iBAT and inguinal WAT (iWAT). Perirenal adipose tissue (PRAT) is bilaterally distributed around the kidney and is encapsulated by a multilayered fibrous membrane, known as the renal fascia (*de Jong et al., 2015*; *Liu et al., 2019*). The medial region of PRAT (herein termed mPRAT) is located around the hilum of the kidney, and several studies have identified UCP1-expressing BAs in this area in human PRAT, suggesting that it represents a form of visceral BAT (*Betz et al., 2013*; *Li et al., 2015*; *Nagano et al., 2015*; *Svensson et al., 2014*; *van den Beukel et al., 2015*; *Vergnes et al., 2016*). In contrast, the lateral region of PRAT (herein termed lPRAT) is predominantly composed of WAs (*de Jong et al., 2015*). Based on morphological changes in adipocytes of human PRAT during aging, a process of continuous replacement of BAs by WAs has been proposed (*Tanuma et al., 1976*; *Tanuma et al., 1975*). On the other hand, analysis of PRAT collected at necropsy from people living in Siberia revealed an up to 40% BA component, indicating the potential of PRAT to respond to low environmental temperature (*Efremova et al., 2020*). Moreover, so-called 'dormant' BAs have been described in PRAT of adult kidney donors, characterized by unilocular morphology, high levels of UCP1 expression, and a gene expression profile that partially overlaps that of WAs (*Jespersen et al., 2019*).

The process by which BAs acquire WA-like properties has been referred to as 'whitening', and it is characterized by the enlargement of lipid droplets and reduced expression of thermogenic genes, such as *Ucp1* (*Shimizu et al., 2014*). Brown-to-white conversion or whitening is a known complication of obesity associated with vascular rarefaction, tissue inflammation, and leptin receptor deficiency (*Kotzbeck et al., 2018*; *Rangel-Azevedo et al., 2022*; *Shimizu et al., 2014*). In addition, warm adaptation, impaired β-adrenergic signaling, and loss of adipose triglyceride lipase have also been implicated in BAT whitening (*Kotzbeck et al., 2018*). Under normal physiological conditions, whitening of BAT has only been demonstrated in rabbit iBAT (*Huang et al., 2022*). With regard to PRAT, it is unclear whether the process by which the number of WAs increases at the expense of BAs during postnatal life reflects direct brown-to-white cell conversion or some other mechanisms of replacement. Moreover, the cellular and molecular properties of the newly formed WA population as well as its spatial distribution in PRAT remain unknown.

In the present study, we took advantage of snRNA-seq and genetic lineage-tracing methods to reveal the existence of a bona fide whitening process of BAs in the mPRAT during mouse postnatal development. We identified a distinct population of newly formed adipocytes derived by whitening, described its anatomical location, and characterized its molecular properties both during its formation and after cold exposure. In addition, our analysis of cells in the lPRAT revealed populations of adipocytes, ASPCs, and macrophages that are transcriptionally distinct from those found in classical visceral fat depots such as epididymal WAT (eWAT).

# Results

## snRNA-seq analysis of murine mPRAT reveals a unique adipocyte subpopulation and uncovers evidence of a brown-to-white transition during postnatal development

To explore the cellular dynamics of mPRAT and iBAT during postnatal development, we collected the tissues from 1-, 2-, and 6-month-old C57BL/6J male mice for whole-tissue snRNA-seq and performed depot-specific analysis. As expected, weight of the collected tissues gradually increased as the animals grew up (*Figure 1—figure supplement 1A*). After strict quality control (*Figure 1—figure supplement 1B and C*), we profiled a total of 11,708 and 31,079 nuclei for mPRAT and iBAT, respectively. In the integrated datasets of mPRAT, we identified six clusters based on differentially expressed genes (DEGs), including adipocyte, ASPC, macrophage, vascular (endothelial cell and pericyte) and mesothelial cells that are commonly present in adipose tissue (*Massier et al., 2023*; *Figure 1A and B*, *Figure 1—figure supplement 1D and E*). mPRAT was mostly composed of adipocytes and the relative proportions of all the cell types remained largely unchanged during the analyzed time points (*Figure 1C*). Similar cell types and cellular composition were identified in the integrated datasets of iBAT, with the exception of the presence of smooth muscle cells and absence of mesothelial cells (*Figure 1D–F*, *Figure 1—figure supplement 1F and G*).

Next, we scrutinized mPRAT adipocyte population in greater detail through further clustering. Adipocytes in murine mPRAT were highly heterogeneous and could be separated into four interconnected subpopulations, all expressing the *Plin1* gene that encodes the lipid droplet protein Perilipin, which we referred to as mPRAT-ad1, 2, 3, and 4, respectively (*Figure 1G and H*, *Figure 1—figure supplement 1H*). mPRAT-ad1 resembled classical BAs and expressed *Ucp1*, *Cidea*, *Pank1*, and *Gk* (*Figure 1H and I*). mPRAT-ad2 was characterized by the expression of *Cidea*, *Pank1*, and *Gk*, but not *Ucp1* (*Figure 1H and I*). In addition, it also expressed detectable levels of *Cyp2e1* and *Slit3* (*Figure 1H and I*), two genes which have recently been implicated in thermogenesis regulation (*Sun et al., 2020*; *Wang et al., 2021*). Such gene expression pattern suggested that mPRAT-ad2 may contain BA-like cells that could be recruited by stimuli that induce thermogenesis. In contrast, mPRAT-ad3, which expressed higher levels of *Cyp2e1* and *Slit3*, lower levels of *Tshr* and *Slc7a10*, but not *Aldh1a1* or *Lep*, and mPRAT-ad4, which expressed *Cyp2e1*, *Slit3*, *Aldh1a1*, *Lep*, and high levels of *Tshr* and *Slc7a10*, more closely resembled classical WAs, of which mPRAT-ad4 would appear to be at a more mature stage (*Figure 1H and I*). Adipocytes of iBAT could also be separated into four distinct subpopulations, which we referred to as iBAT-ad1, 2, 3, and 4, respectively (*Figure 1—figure supplement 1I*). iBAT-ad1, 2, and 3 contained BAs that expressed different levels of *Ucp1* and *Cidea*, and may correspond to previously identified low- and high-thermogenic BA subpopulations (*Song et al., 2020*; *Figure 1—figure supplement 1J–L*). Altogether, they comprised over 90% of the total adipocyte population in iBAT (*Figure 1—figure supplement 1I*). On the other hand, iBAT-ad4 expressed *Cyp2e1*, *Slit3*, *Slc7a10*, *Aldh1a1*, and *Lep*, and module score analysis of signature genes showed it to bear close similarity to the recently identified P4 adipocyte subpopulation that regulates thermogenesis in iBAT (*Sun et al., 2020*; *Figure 1—figure supplement 1M*). Comparison between mPRAT and iBAT adipocyte transcriptomes using module score analysis revealed that mPRAT-ad1 was more similar to all iBAT BA subpopulations (iBAT-ad1, 2, and 3), while mPRAT-ad3 and 4 were more similar to iBAT-ad4 (*Figure 1—figure supplement 1N*). Interestingly, we could not find a match for mPRAT-ad2 among iBAT adipocytes (*Figure 1—figure supplement 1N*), suggesting that it represents a distinct subpopulation of adipocytes specific to mPRAT, but absent in classical BAT.

Closer inspection of mPRAT adipocyte composition at 1, 2, and 6 mo revealed a gradual decrease in the proportions of mPRAT subpopulations ad1 (43.0, 34.9, and 33.1%, respectively) and ad2 (36.1, 31.2, and 23.9%, respectively), which was accompanied by a reciprocal increase in ad3 (14.7, 26.8, and 29.9%, respectively) and ad4 (6.2, 7.1, and 13.1%, respectively) (*Figure 1G*), suggesting the existence of a progressive transition in the adipocyte composition of mPRAT during postnatal life. This possibility was further investigated by Monocle 3 trajectory analysis of these cells, which revealed a single continuous trajectory linking the four subpopulations (*Figure 1J*). Based on the pattern of composition changes observed through time (*Figure 1G*), we defined mPRAT-ad1 subpopulation as the starting point of the trajectory uncovering pseudo-time values that increased steadily from mPRAT-ad1 to mPRAT-ad4 (*Figure 1J and K*). Along the pseudo-timeline, BA module genes such as *Ucp1*, *Cidea*,

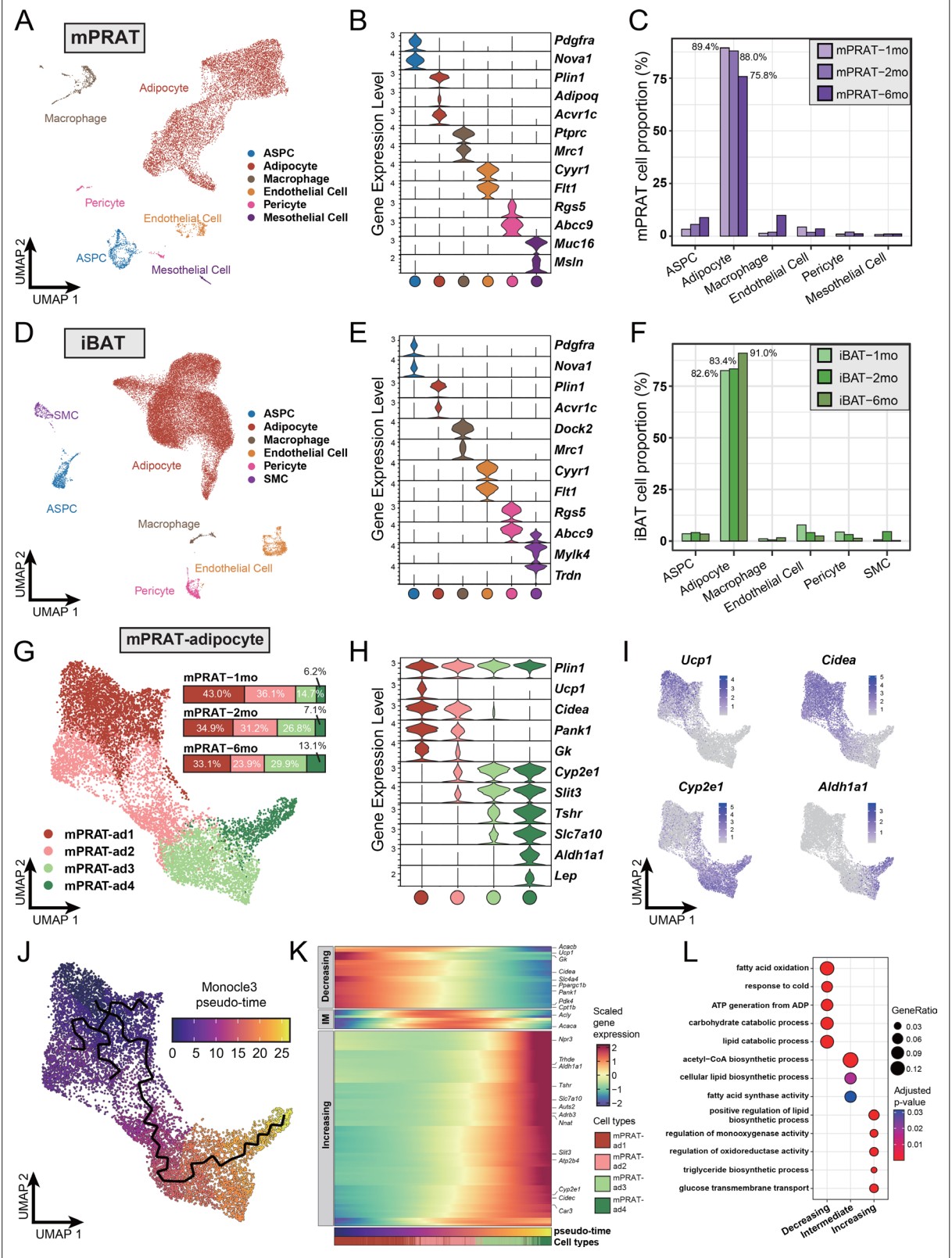

**Figure 1.** Single-nucleus RNA sequencing (snRNA-seq) reveals a unique adipocyte subpopulation and delineates a brown-to-white transition process in the medial region of perirenal adipose tissue (mPRAT) during postnatal development. (**A**) Uniform Manifold Approximation and Projection (UMAP) of all cell types in mPRAT from 1-, 2-, and 6-month-old C57BL/6J male mice. A total of 11,708 nuclei were integrated, including 3342, 3608, and 4758 nuclei from 15, 12, and 8 animals for the three time points, respectively. (**B**) Violin plot of the marker gene expression levels of all the identified cell types in

*Figure 1 continued on next page*

*Figure 1 continued*

mPRAT. Expression level was log normalized for all the violin plots in the study. Cell types are represented by circles following the same color scheme in corresponding UMAP. (**C**) Histogram illustrating the percentage of each cell type in mPRAT relative to the total number of analyzed nuclei. (**D**) UMAP of all cell types in interscapular brown adipose tissue (iBAT) from 1-, 2-, and 6-month-old C57BL/6J male mice. A total of 31,079 nuclei were integrated, including 6468, 8323, and 16,288 nuclei from the same mice littler as that in mPRAT for the three time points, respectively. SMC, smooth muscle cell. (**E**) Violin plot of the marker gene expression levels of all the identified cell types in iBAT. (**F**) Histogram illustrating the percentage of each cell type in iBAT relative to the total number of analyzed nuclei. (**G**) UMAP and cellular composition of the adipocyte subpopulations in mPRAT from three time points. (**H**) Violin plot of the marker gene expression levels of each mPRAT adipocyte subpopulation. (**I**) *Ucp1, Cidea, Cyp2e1,* and *Aldh1a1* expression pattern in the mPRAT adipocyte subpopulations. Scale bar represents log normalized gene expression levels. (**J**) Monocle 3 trajectory map of the mPRAT adipocyte subpopulations. (**K**) Heat map of differentially expressed genes (DEGs) along the pseudo-time trajectory in (**J**). Genes were selected by Moran's I test in Monocle 3 with a threshold larger than 0.2. Representative genes were highlighted. Adipocytes were aligned along the increasing pseudo-timeline and plotted using Z-scaled gene expression. Gene module patterns, including decreasing, intermediate (IM), and increasing, were determined using k-means clustering (k = 3). (**L**) Gene ontology (GO) analysis of DEGs from each module in (**K**). Color and size of each circle represent the adjusted p-value by Benjamini and Hochberg method and the gene ratio within each module, respectively.

The online version of this article includes the following figure supplement(s) for figure 1:

**Figure supplement 1.** Tissue weight, quality control, and additional analysis on the single-nucleus RNA sequencing (snRNA-seq) data obtained from the medial region of perirenal adipose tissue (mPRAT) and interscapular brown adipose tissue (iBAT) of 1-, 2-, and 6-month-old C57BL/6J male mice.

**Figure supplement 2.** Analysis on the adipose stem and progenitor cell (ASPC) population of the medial region of perirenal adipose tissue (mPRAT) and interscapular brown adipose tissue (iBAT) and quality control of the lateral region of PRAT (lPRAT) datasets.

*Ppargc1b*, *Gk,* and *Pank1* showed decreasing expression levels, whereas WA module genes such as *Npr3*, *Aldh1a1*, *Tshr*, *Slc7a10*, *Slit3*, and *Cyp2e1* showed the opposite trend (***Figure 1K***). Gene ontology (GO) enrichment analysis of these genes revealed a progressive transition from catabolic to anabolic processes along the pseudo-time trajectory (***Figure 1L***). The midpoint of the trajectory, corresponding to the mPRAT-ad2 subpopulation, coincided with the highest expression levels of lipogenic genes, including *Acly* and *Acaca*, as well as enrichment in several synthetic pathways (***Figure 1K and L***). A similar analysis on iBAT adipocytes failed to reveal a distinct trajectory and its pseudo-time length of up to 12.8 (***Figure 1—figure supplement 1O***) was much shorter than that found in mPRAT adipocytes, which reached 26.7 (***Figure 1J***), suggesting a less heterogeneous and more stable adipocyte population in iBAT. Taken together, these results suggested the occurrence of a brown-to-white transition during postnatal development in adipocytes of mouse mPRAT.

Re-clustering of mPRAT ASPC population revealed three subpopulations (referred to as mPRAT-aspc1 to 3, respectively) that shared the expression of the ASPC marker gene *Pdgfra* (***Figure 1—figure supplement 2A–C***). mPRAT-aspc1 (expressing *Dpp4*, *Pi16*, *Anxa3*) was identified as fibro-adipogenic progenitors (FAPs), while mPRAT-aspc2 (expressing *Pdgfrb* and low levels of *Pparg*) and mPRAT-aspc3 (expressing high levels of *Pparg*, *Cd36*, *Cidea*) represented preadipocytes at different stages of differentiation (***Figure 1—figure supplement 2B***). mPRAT-aspc1 and 2 comprised the majority of the ASPC population at all three time points and a reciprocal switch in their relative proportions was observed between 2 and 6 months of age (***Figure 1—figure supplement 2D***). iBAT ASPCs could also be re-clustered into three distinct subpopulations (termed iBAT-aspc1 to 3, respectively), which presented good correspondence to the three ASPCs subpopulations identified in mPRAT (***Figure 1—figure supplement 2E–G***). However, unlike mPRAT, iBAT ASPC subpopulations showed more dynamic changes in their relative composition during the three postnatal times examined (***Figure 1—figure supplement 2H***). We further investigated the apparent similarities between mPRAT and iBAT ASPCs by co-clustering the two cell populations. This revealed a high degree of overlap between them with similar marker genes and good correspondence among their different subpopulations (***Figure 1—figure supplement 2I–K***), indicating similar transcriptomes and suggesting that they may give rise to similar types of adipocytes. Thus, the fact that PRAT-specific mPRAT-ad2 is absent in iBAT suggested to us that those cells may arise from brown-to-white conversion of adipocyte subpopulations rather than ASPC differentiation.

## The cellular composition of lPRAT is significantly different from that in eWAT at the transcriptional level

Next, we explored the transcriptome of lPRAT in 2- and 6-month-old C57BL/6J male mice using snRNA-seq. The integrated datasets comprised a total of 6048 nuclei after filtering (***Figure 1—figure***

*supplement 2L*) and allowed us to identify four distinct clusters, including adipocytes, ASPCs, macrophages, and mesothelial cells (*Figure 2A and B*). The proportion of adipocytes in lPRAT decreased from 2 to 6 mo (70.6% to 46.5%), accompanied by a reciprocal increase in the macrophage population (5.7% to 23.8%) (*Figure 2C*). Such changes have previously been shown to mark an aging signature in visceral adipose tissue (*Lumeng et al., 2011*). As expected, lPRAT adipocytes were enriched in WA (*Lep* and *Acvr1c*) but not BA markers (*Ucp1* and *Cidea*) (*Figure 2B*).

Re-clustering of the adipocyte population revealed four distinct subpopulations (lPRAT-ad1 to 4) (*Figure 2D*), all of which expressed high levels of genes typical of WAT such as *Adrb3*, *Lpl*, and *Cd36* (*Figure 2E*). lPRAT-ad1 expressed genes related to lipogenesis (*Fasn, Acly*), endocytosis (*Ighm, Anxa2*), and stress response (*Fgf1, Hif1a*) (*Figure 2E*). lPRAT-ad-2 comprised more than half of the total lPRAT adipocyte population (*Figure 2D*) and expressed high levels of genes that are involved in adipogenesis regulation, such as *Irf2* and *Mitf* (*Figure 2E*). lPRAT-ad3 and 4 expressed genes that oppose lipogenesis (*Cdk8*) and control lipid handling (*Camk1d*) (*Figure 2E*). Reciprocal changes in the relative proportions of lPRAT-ad3 and lPRAT-ad4 were observed from 2 to 6 mo: lPRAT-ad3 increased from 4.7% to 17.2% at the expense of lPRAT-ad4 that decreased from 21.5% to 4.1% (*Figure 2D*). We then compared the transcriptome of lPRAT adipocytes with that of eWAT, a better characterized visceral fat depot, using a previously published eWAT snRNA-seq dataset (*Sárvári et al., 2021*). Unexpectedly, the two populations showed only a partial overlap (*Figure 2F*). Specifically, lPRAT-ad1 resembled the lipogenic adipocyte and the stressed lipid scavenging adipocyte subpopulations previously described in eWAT, while lPRAT-ad2 was more similar to the lipid-scavenging adipocyte subpopulation of eWAT. In contrast, however, lPRAT-ad3 and 4 showed no overlap with previously identified subpopulations of eWAT adipocytes (*Figure 2F*), suggesting that they may represent unique cell types to the lPRAT depot.

The ASPC population of lPRAT could be subclustered into two subpopulations: FAPs characterized by expression of *Dpp4* and *Pi16* (lPRAT-aspc1) and preadipocytes expressing *Pdgfrb*, *Pparg*, and *Cd36* (lPRAT-aspc2) that comprised over 85% of the ASPC population at both time points (*Figure 2G and H*). Co-clustering of the lPRAT APSC population with the one previously defined in eWAT by Sarvari et al. revealed a partial overlap with their FAP2 and FAP3 subpopulations, although the bulk of lPRAT ASPCs appeared largely unrelated to ASPCs from eWAT (*Figure 2I*). Sub-clustering of the macrophages in lPRAT identified three subpopulations (lPRAT-mac1 to 3), all of which expressed typical macrophage markers, including *Adgre1* and *Mctp1* (*Figure 2J–K*). lPRAT-mac1 specifically expressed *Trem2* and *Atp6v0d2*, signature macrophage genes that have recently been proposed to participate in the maintenance of metabolic homeostasis under high-fat diet (*Dai et al., 2022*; *Jaitin et al., 2019*; *Figure 2K*). Intriguingly, this subpopulation increased almost tenfold (from 8.1% to 73.8%) from 2 to 6 mo (*Figure 2J*), in line with the notion that aging mimics some of the gene signatures induced by high-fat diet (*Lumeng et al., 2011*). A reciprocal decrease in proportion of both lPRAT-mac2, characterized by *Fn1* expression, and lPRAT-mac3, marked by expression of *Cd163* and *Plekhg5,* was also found during this period of time (*Figure 2J and K*). Unexpectedly, comparison with the macrophage subpopulations identified in eWAT by Sarvari et el. revealed that lPRAT and eWAT macrophage populations were very different from each other (*Figure 2L*), suggesting dissimilar local microenvironments in the two tissues (*Lavin et al., 2014*). Taken together, these results indicated that the cellular composition of lPRAT is significantly different from that of eWAT, despite both being visceral WAT depots.

## Evidence of brown-to-white adipocyte conversion in the periureter region of mPRAT

UCP1 immunostaining in kidney explants and attached PRAT from juvenile (6 wk) and adult (8 mo) mice was more intense in the anterior portion of the mPRAT that wraps around the renal blood vessels (herein termed perivascular PRAT [pvPRAT]) compared to the posterior region covering the ureter (herein termed periureter PRAT [puPRAT]) (*Figure 3—figure supplement 1A and B*), suggesting an uneven distribution of UCP1-expressing BAs in mPRAT. In addition, UCP1 staining was decreased in adult compared to juvenile mPRAT and absent from lPRAT, as expected (*Figure 3—figure supplement 1A*). In order to investigate brown-to-white conversion of mPRAT adipocytes during postnatal life, we used the *Ucp1Cre;*Ai14 mouse line that drives expression of tdTomato in BAs from the moment these activate the *Ucp1* gene (*Kong et al., 2014*). UCP1 immunostaining of mPRAT sections in 2-month-old *Ucp1Cre;*Ai14 mice enabled us to assess whether and to which extent tdTomato⁺ adipocytes had lost

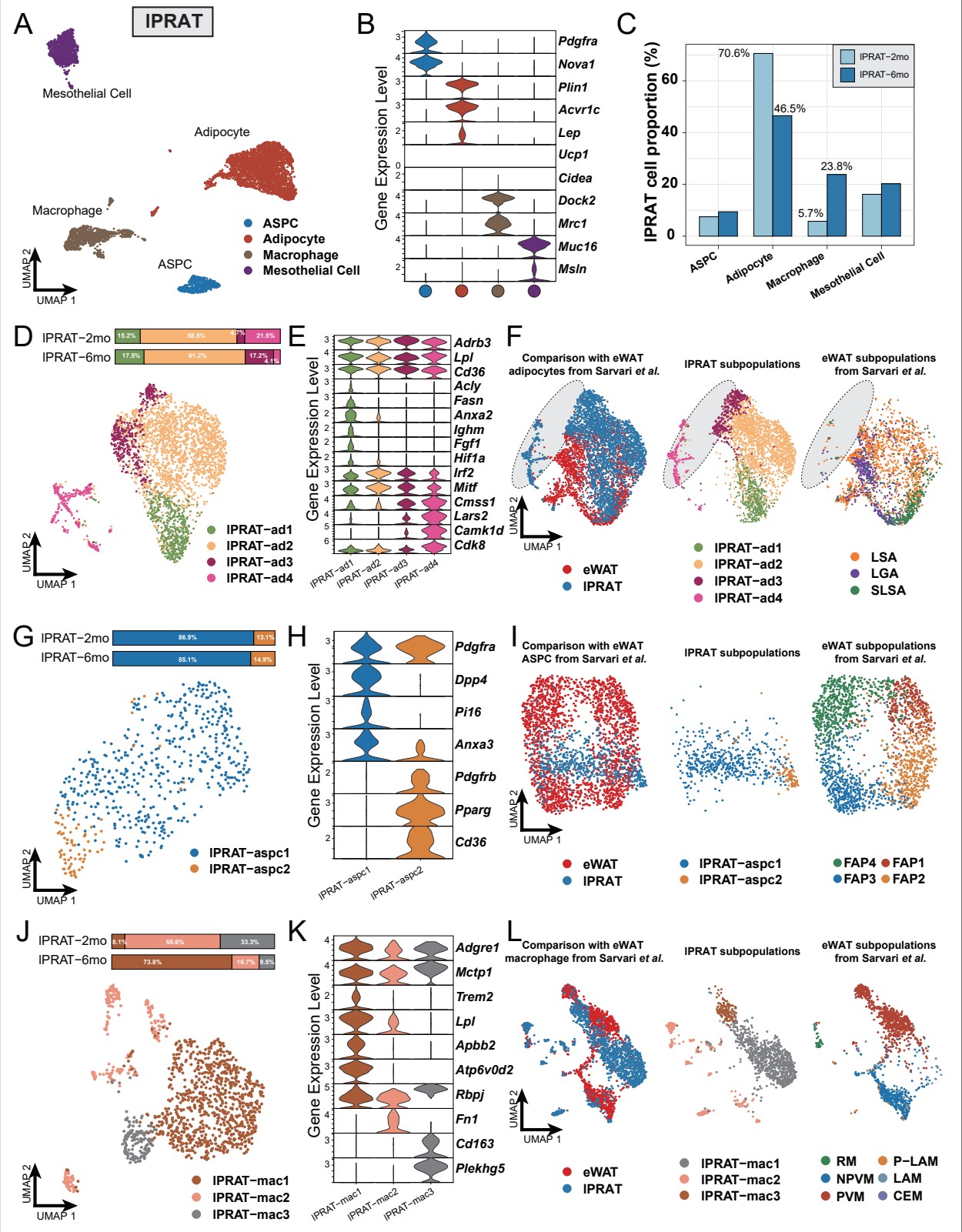

**Figure 2.** Lateral region of perirenal adipose tissue (lPRAT) has distinct cellular composition compared with epididymal white adipose tissue (eWAT). (**A**) Uniform Manifold Approximation and Projection (UMAP) of all cell types in lPRAT from 2- and 6-month-old C57BL/6J male mice. A total of 6048 nuclei were analyzed, including 1428 and 4620 nuclei for the two time points, respectively. (**B**) Violin plot of the marker gene expression levels of all cell types in lPRAT. (**C**) Histogram illustrating the percentage of each cell type in lPRAT relative to the total number of analyzed nuclei. (**D, G, J**) UMAP and

*Figure 2 continued on next page*

*Figure 2 continued*

cellular composition of the adipocyte (**D**), adipose stem and progenitor cell (ASPC) (**G**), and macrophage (**J**) subpopulations in lPRAT. (**E, H, K**) Violin plot of the marker gene expression levels of each adipocyte (**E**), ASPC (**H**), and macrophage (**K**) subpopulation in lPRAT. (**F, I, L**) Merged clusters of lPRAT adipocytes (**F**), ASPCs (**I**), and macrophages (**L**) with the corresponding clusters in *Sárvári et al., 2021*. Gray oval shape in (**F**) highlights the unique adipocyte subpopulation in lPRAT. LSA, lipid-scavenging adipocyte; LGA, lipogenic adipocyte; SLSA, stressed lipid-scavenging adipocyte. RM, regulatory macrophage; NPVM, non-perivascular macrophage; PVM, perivascular macrophage; LAM, lipid-associated macrophage; P-LAM, proliferating LAM; CEM, collagen-expressing macrophage.

UCP1 expression, an indication of brown-to-white conversion (*Figure 3A*). This analysis revealed that 61.8% of tdTomato$^+$ cells in puPRAT no longer expressed UCP1 by 2 months of age (*Figure 3A and B*). In contrast, only 7.5% of tdTomato$^+$ cells did not express UCP1 in pvPRAT at this age (*Figure 3A and B*). These results suggested a significant loss of the BA phenotype in puPRAT, consistent with brown-to-white conversion in this region. In agreement with this, many tdTomato$^+$&UCP1$^-$ adipocytes displayed one large unilocular lipid droplet, a morphological hallmark of WAs (insets in *Figure 3A*). In line with our analysis of kidney explants, puPRAT contained a significantly lower proportion of tdTomato$^+$ adipocytes (*Figure 3C*) and adipocytes that expressed UCP1 at 2 month of age (UCP1$^+$) (*Figure 3D*) compared to pvPRAT. A similar analysis of iBAT failed to reveal evidence of brown-to-white conversion (*Figure 3A and B*), in agreement with previous studies (*Huang et al., 2022*). Histological analysis of tissue sections of mouse PRAT and iBAT from 1 to 6 months old revealed that the size of lipid droplets increased by around 5.5-fold in puPRAT and about 2.3-fold in pvPRAT (*Figure 3E and F*), in agreement with a more pronounced brown-to-white transition in the puPRAT region. Immunostaining of the puPRAT region for CYP2E1 and ALDH1A1, markers of mPRAT-ad3/4 and -ad4 subpopulations of mPRAT WAs, respectively, revealed that a proportion of tdTomato$^+$&UCP1$^-$ adipocytes expressed these genes (34 and 23%, respectively, *Figure 3G–I*). This indicated that cells that are initially BAs can undergo brown-to-white conversion all the way to mPRAT-ad3 and -ad4 WA subtypes, as predicted by our pseudo-time trajectory analysis. The remaining fraction of tdTomato$^+$&UCP1$^-$ adipocytes have likely transitioned to the mPRAT-ad2 subtype. To visualize the mPRAT-ad2 adipocytes, we collected pvPRAT and puPRAT from the *Ucp1CreERT2;*Ai14 mice 1 d after the last tamoxifen injection and stained with CYP2E1 antibody and BODIPY. With over 93% specific labeling rates of the *Ucp1CreERT2* (*Figure 3—figure supplement 1C and D*), we revealed a significantly higher percentage of mPRAT-ad2 cells (Tomato$^-$&CYP2E1$^-$&BODIPY$^+$) in puPRAT compared with pvPRAT (*Figure 3—figure supplement 1E*).

Together, these results pointed to the existence of a subpopulation of BAs, the majority of which resided in puPRAT, that underwent brown-to-white conversion during postnatal growth, characterized by loss of UCP1 and increase in lipid droplet size, to become mPRAT-ad3 and 4 subpopulations of WAs as well as the more intermediate subpopulation of mPRAT-ad2.

## Cold exposure prevents brown-to-white conversion and induces UCP1 expression in mPRAT

A significant increase in UCP1 mRNA and protein levels was observed in mPRAT and iBAT of 1-month-old mice following 4-week cold exposure (2 wk at 18°C + 2 wk at 10°C), but UCP1 expression could not be detected in lPRAT under either condition (*Figure 4A–C*). Cold exposure had a dramatic effect on lipid droplet size in pvPRAT and iBAT, but less so in puPRAT, which retained several large lipid droplets even after cold exposure (*Figure 4D*). These results indicated that mPRAT, especially the perivascular region, can indeed respond to cold exposure in a fashion similar to canonical BAT from iBAT. Not only did cold exposure enhanced the thermogenic phenotype of mPRAT, but it also counteracted brown-to-white conversion of BAs in this tissue. Following cold exposure, the proportion of tdTomato$^+$&UCP1$^-$ adipocytes in *Ucp1CreERT2;*Ai14 mice was significantly lower compared to room temperature in both puPRAT and pvPRAT (*Figure 4E–G*). Interestingly, the proportion of tdTomato$^-$&UCP1$^+$ adipocytes was increased after cold exposure in both areas (*Figure 4H*). This population of cold-recruited BAs might have arisen de novo through differentiation of ASPCs and/or conversion from UCP1$^-$ adipocytes. Using *PdgfraCreERT2;*Ai14 mice subjected to the same experimental conditions, we found that cold exposure increased the proportion of tdTomato$^+$&UCP1$^+$ adipocytes (namely ASPC-derived BAs) in puPRAT by a much smaller magnitude (2.3-fold) than the total increase in cold-recruited BAs (10- fold), while the difference was less dramatic in pvPRAT (4.4- vs 2.2-fold) (*Figure 4H–K*). Adipocytes in pvPRAT and puPRAT of the *PdgfraCre;*Ai14 mice were 100%

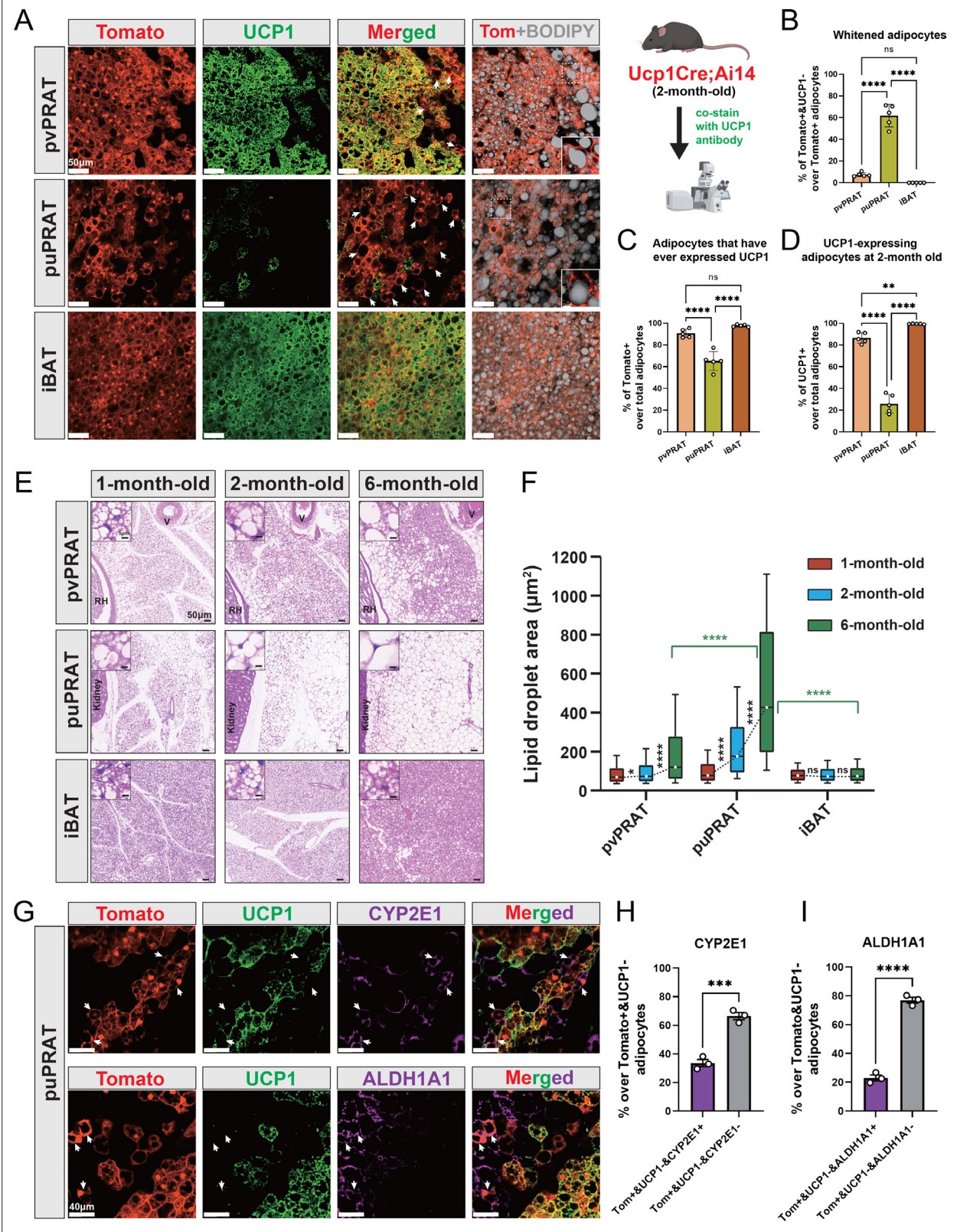

**Figure 3.** Periureter perirenal adipose tissue (puPRAT) contains majority of the whitened brown adipocytes (BAs). (**A**) Representative immunofluorescence images of tdTomato (red) and UCP1 (green) expression in perivascular PRAT (pvPRAT) , puPRAT, and interscapular brown adipose tissue (iBAT) of 2-month-old *Ucp1Cre;*Ai14 male mice. Whitened adipocytes are marked with Tomato expression, but not UCP1 (white arrows). Lipid droplets were stained with BODIPY (gray). Insets highlight the whitened adipocytes with relatively large lipid droplet size. Scale bar, 50 μm. (**B–**

*Figure 3 continued on next page*

*Figure 3 continued*

**D**) Quantification of the percentage of Tomato⁺&UCP1⁻ (**B**), Tomato⁺ (**C**), and UCP1⁺ (**D**) cells in (**A**). n = 5 mice for each tissue, represented by a dot in the graph. Three tissue slices were quantified for each mouse for all analysis in the study. Bars represent mean± SD for all analysis in the study. **p<0.01; ****p<0.0001 by one-way ANOVA. (**E**) Representative hematoxylin and eosin (HE) images of pvPRAT, puPRAT, and iBAT of 1-, 2-, and 6-month-old C57BL/6J male mice. Scale bar, 50 μm. V, blood vessel; RH, renal hilum. (**F**) Quantification of the lipid droplet size in (**E**). A total of 3082–4857 adipocytes from 4 to 6 mice were quantified for each tissue. Boxplot shows the area distribution of all lipid droplets where the whiskers show the 10–90 percentile. *p<0.05; ****p<0.0001 by two-way ANOVA. (**G**) Representative immunofluorescence images of tdTomato (red), UCP1 (green), and CYP2E1/ALDH1A1 (magenta) expression in puPRAT of 2-month-old *Ucp1Cre;*Ai14 male mice. Tomato⁺&UCP1⁻&CYP2E1⁺ and Tomato⁺&UCP1⁻&ALDH1A1⁺ cells represent the whitened adipocytes that became mPRAT-ad3 and 4 adipocytes (white arrows). Scale bar, 40 μm. (**H and I**) Quantification of the percentage of Tomato⁺&UCP1⁻&CYP2E1⁺ (**H**) and Tomato⁺&UCP1⁻&ALDH1A1⁺ cells (**I**) in (**G**). n = 3 mice for each tissue, represented by a dot in the graph. Bars represent mean ± SD. ***p<0.001; ****p<0.0001 by unpaired *t*-test.

The online version of this article includes the following figure supplement(s) for figure 3:

**Figure supplement 1.** Difference in adipocyte composition between perivascular perirenal adipose tissue (pvPRAT) and periureter PRAT (puPRAT).

tdTomato⁺, indicating that they were entirely derived from *Pdgfra*-expressing cells (***Figure 4—figure supplement 1***). These results indicated that new BAs induced by cold exposure were mainly derived from UCP1⁻ adipocytes rather than de novo ASPC differentiation in puPRAT (***Figure 4L***), whereas the two processes contributed equally in the perivascular region. Cold exposure also affected brown-to-white conversion in adult animals, as shown by a significant reduction in the proportion of tdTomato⁺&UCP1⁻ adipocytes in both puPRAT and pvPRAT (***Figure 5A–C***). Similar increase in the proportion of tdTomato⁻&UCP1⁺ adipocytes after cold exposure was also observed in both areas (highlighted by yellow arrows in ***Figure 5B***).

## mPRAT-ad2 is the major adipocyte subpopulation induced by cold exposure to become BA in adult mPRAT

In order to reveal the first line of response to cold exposure and identify the main affected subpopulation among mPRAT adipocytes, we subjected 6-month-old mice to a more acute cold exposure (4°C for 3 d) and collected the mPRAT for snRNA-seq analysis. As expected, cold exposure caused a dramatic decrease in lipid droplet size in pvPRAT, puPRAT, iBAT, and iWAT (***Figure 5—figure supplement 1A***). After filtering (***Figure 5—figure supplement 1B***), a total of 7387 nuclei could be clustered into adipocytes, ASPCs, macrophages, vascular cells (endothelial cell and pericyte), and mesothelial cells (***Figure 5—figure supplement 1C and E***), without apparent difference in the relative proportions of these cell types between mice exposed to cold and those kept at room temperature (***Figure 5—figure supplement 1D***). Upon subclustering, the adipocyte population was separated into four subpopulations (***Figure 5D–F***) that were highly similar to the four mPRAT adipocyte subpopulations identified in our previous datasets (mPRAT-ad1-4) (***Figure 5—figure supplement 1F***). Interestingly, cold exposure had dramatic effects on the composition of mPRAT: the proportion of mPRAT-ad1 doubled, the mPRAT-ad2 cells all disappeared, the mPRAT-ad3 subpopulation was reduced by 30%, while the proportion of mPRAT-ad4 cells remained unaffected (***Figure 5E***). As no enhanced cell death was observed upon cold exposure (***Figure 5—figure supplement 1G***), it is possible that the entire mPRAT-ad2 and part of the mPRAT-ad3 subpopulations were converted to the thermogenic *Ucp1*⁺ mPRAT-ad1 subpopulation upon a dramatic increase in *Ucp1* expression following cold exposure (***Figure 5F***). Thus, the mPRAT-ad2 subpopulation that underwent brown-to-white conversion during postnatal life would appear to be the major subpopulation that reverted to a thermogenic, *Ucp1*⁺ phenotype upon cold exposure in adult mPRAT. Comparison of differentially enriched genes and gene regulatory networks (GRNs) in mPRAT-ad1, 3, and 4 after cold exposure or at room temperature is presented in ***Figure 5G–J***. We also noted that the proportions of ASPC subpopulations in mPRAT remained relatively unchanged after cold exposure (***Figure 5—figure supplement 1H and I***), suggesting that precursor cells may be less sensitive to acute environmental temperature drop and have a limited contribution to mature adipocyte phenotypes through de novo adipogenesis after cold exposure.

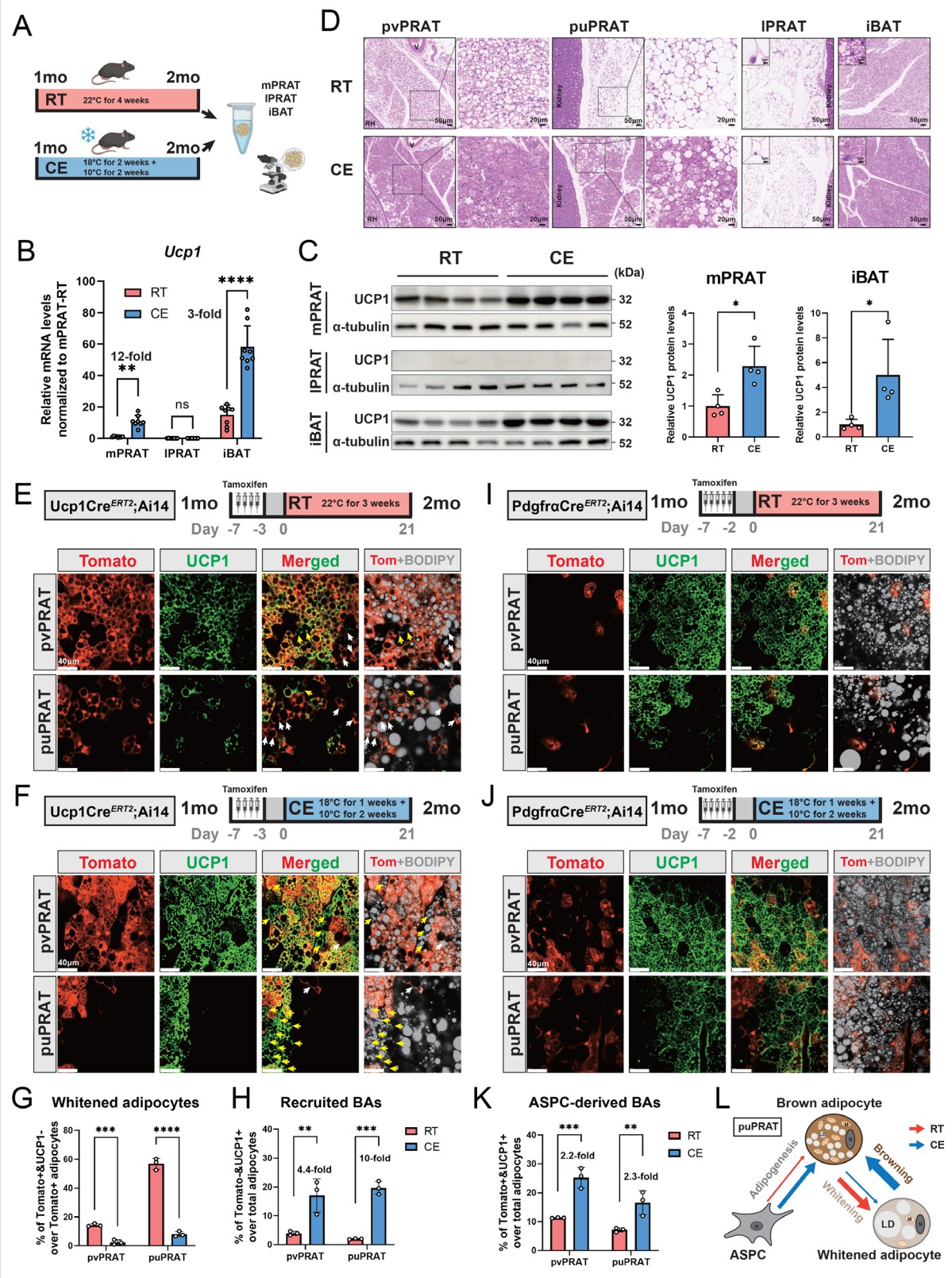

**Figure 4.** Cold exposure inhibits brown adipocyte (BA) whitening and restores UCP1 expression in UCP1⁻ adipocytes of the medial region of perirenal adipose tissue (mPRAT). (**A**) Illustration of the cold exposure (CE) experimental design. (**B**) qPCR analysis of *Ucp1* mRNA level in mPRAT, lateral region of PRAT (lPRAT), and interscapular brown adipose tissue (iBAT) of 2-month-old C57BL/6J male mice in (**A**). n = 8 mice for each condition, represented by a dot in the graph. **p<0.01; ****p<0.0001 by one-way ANOVA. (**C**) Western blot and quantification of UCP1 protein level in mPRAT and iBAT of

*Figure 4 continued on next page*

*Figure 4 continued*

2-month-old C57BL/6J male mice in (**A**). n = 4 mice for each condition, represented by a dot in the graph. *p<0.05 by unpaired *t*-test. (**D**) Representative HE images of perivascular PRAT (pvPRAT), periureter PRAT (puPRAT), lPRAT, and iBAT of 2-month-old C57BL/6J male mice in (**A**). Insets highlight the changes in lipid droplet size. Scale bar, 50 μm. (**E, F**) Representative immunofluorescence images of tdTomato (red) and UCP1 (green) expression in pvPRAT and puPRAT of 2-month-old *Ucp1CreERT2*;Ai14 male mice kept under room temperature (RT) (**E**) or CE (**F**) condition. Tamoxifen was injected at 1-month-old to trace the UCP1-expressing cells. Tomato$^+$&UCP1$^-$ cells represent the population that is whitened during the 1- to 2-month-old tracing period (white arrows). Tomato$^-$&UCP1$^+$ cells represent the recruited BAs (yellow arrows). Scale bar, 40 μm. (**G, H**) Quantification of the percentage of whitened adipocytes (Tomato$^+$&UCP1$^-$) (**G**) and recruited BAs (Tomato$^-$&UCP1$^+$) (**H**) in (**E, F**). n = 3 mice for each tissue, represented by a dot in the graph. **p<0.01; ***p<0.001; ****p<0.0001 by two-way ANOVA. (**I, J**) Representative immunofluorescence images of tdTomato (red) and UCP1 (green) expression in pvPRAT and puPRAT of 2-month-old *PdgfraCreERT2*;Ai14 male mice kept under RT (**I**) or CE (**J**) condition. Tamoxifen was injected at 1-month-old to trace the Pdgfra-expressing cells. Scale bar, 40 μm. (**K**) Quantification of the percentage of adipose stem and progenitor cell (ASPC)-derived BAs (Tomato$^+$&UCP1$^+$) in (**I, J**). n = 3 mice for each tissue, represented by a dot in the graph. **p<0.01; ***p<0.001 by two-way ANOVA. (**L**) Illustration of the differential cellular contributions to the cold-recruited adipocytes in puPRAT. CE activated ASPC-derived BA adipogenesis and browning of whitened adipocytes, while preventing BA whitening. N, nucleus; M, mitochondria; LD, lipid droplet.

The online version of this article includes the following source data and figure supplement(s) for figure 4:

**Source data 1.** Original files of the full raw unedited blots of *Figure 4C*.

**Source data 2.** Figures with the uncropped blots of *Figure 4C*.

**Figure supplement 1.** Medial region of perirenal adipose tissue (mPRAT) adipocytes are entirely derived from Pdgfra-expressing cells.

## mPRAT-ad2 subtype is transcriptionally different from iWAT adipocytes under both room temperature and cold exposure conditions

Having demonstrated the existence of an adipocyte subpopulation in mPRAT that can be induced to express a thermogenic profile in response to cold exposure, we sought to perform a comparison with the transcriptional profiles of iBAT and iWAT adipocytes subjected to the same conditions in the same animals. In iBAT, analysis of 27,717 nuclei after filtering (*Figure 6—figure supplement 1A*) revealed no obvious differences in the proportions of all cell types (*Figure 6—figure supplement 1B–D*) or ASPC subpopulations (*Figure 6—figure supplement 1H and I*) between room temperature and cold exposure. Following cold exposure, all adipocyte subpopulations showed similar transcriptomes with those identified in the longitudinal datasets (*Figure 6—figure supplement 1G*) and experienced a dramatic increase in the expression levels of BA marker genes, including iBAT-ad4, which had not shown expression for any of those genes at room temperature (*Figure 6—figure supplement 1F*). Proportion of the iBAT-ad1 subpopulation was increased while that of iBAT-ad3 decreased (*Figure 6—figure supplement 1E*), indicating that cold exposure can induce cell-type conversions among *Ucp1*$^+$ adipocyte subpopulations in iBAT, in agreement with previous observations (*Song et al., 2020*). In iWAT, analysis of 11,486 nuclei revealed clusters of adipocytes, ASPCs, immune cells (macrophage, B cell, T cell, neutrophil and dendritic cell), and vascular cells (endothelial cell and pericyte), whose relative proportions remained unchanged after cold exposure (*Figure 6—figure supplement 2A–E*). Five subclusters were identified within the adipocyte population, herein termed iWAT-ad1 to 5 (*Figure 6—figure supplement 2F–H*). As expected, the adipocytes expressing high levels of *Ucp1* and *Cidea* (iWAT-ad1, namely BeAs) were only present in animals that were exposed to cold (*Figure 6—figure supplement 2G*). Surprisingly, a second adipocyte subpopulation that was almost specific to cold-exposed animals (iWAT-ad2) did not express any of the common thermogenic markers, but instead specifically expressed genes involved in lipogenesis (*Acss2, Fasn,* and *Acly*) (*Figure 6—figure supplement 2G and H*). This is in agreement with a recent study showing that expression of lipogenic genes is highly induced in iWAT adipocytes upon cold exposure (*Holman et al., 2023*). In contrast to iWAT-ad1 and ad2, the abundance of iWAT-ad4 cells decreased by half following cold exposure (*Figure 6—figure supplement 2G*). Similar to mPRAT and iBAT, no differences were detected in the proportions of different iWAT ASPCs subtypes following cold exposure (*Figure 6—figure supplement 2I and J*).

Surprisingly, little overlap, if at all, could be found between subpopulations of mPRAT and iWAT adipocytes in mice housed at room temperature (*Figure 6A and B*). In fact, the mPRAT and iWAT adipocytes were mostly exclusive from each other, besides the slight overlap between the two mature WA subpopulations (mPRAT-ad4 and iWAT-ad5) (*Figure 6A and B*). Thus, the cold-inducible subpopulation of mPRAT-ad2 adipocytes would appear to be quite distinct from adipocyte subpopulations found in either iBAT or iWAT under room temperature condition. Upon cold exposure, BeAs in iWAT (iWAT-ad1) showed a good correspondence with a fraction of the mPRAT-ad1 subpopulation of BAs

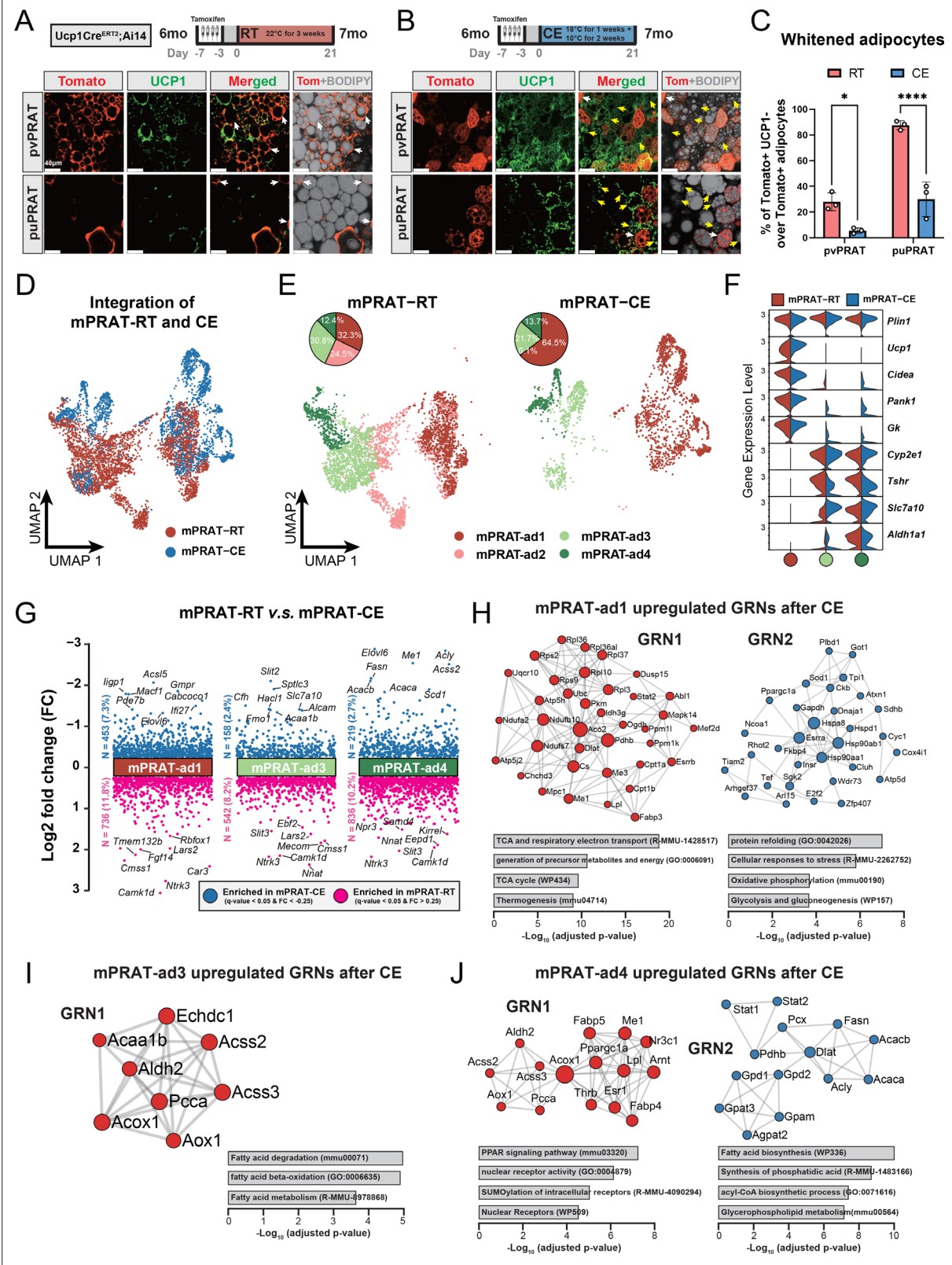

**Figure 5.** mPRAT-ad2 is the major cold-recruitable adipocyte in the medial region of perirenal adipose tissue (mPRAT). (**A, B**) Representative immunofluorescence images of tdTomato (red) and UCP1 (green) expression in perivascular PRAT (pvPRAT) and periureter PRAT (puPRAT) of 7-month-old *Ucp1CreERT2*;Ai14 male mice. Tamoxifen was injected at 6 months old to trace the UCP1-expressing cells. Tomato+&UCP1- cells represent the population that is whitened during the 6- to 7-month-old tracing period (white arrows). Tomato-&UCP1+ cells represent the recruited brown adipocytes

*Figure 5 continued*

(BAs) (yellow arrows). Scale bar, 40 µm. (**C**) Quantification of the percentage of whitened BAs (Tomato$^+$&UCP1$^-$) in (**A, B**). n = 3 mice for each tissue, represented by a dot in the graph. *p<0.05; ****p<0.0001 by two-way ANOVA. (**D, E**) Integrated (**D**) and separated (**E**) Uniform Manifold Approximation and Projection (UMAP) of all cell types in mPRAT of 6-month-old C57BL/6J male mice kept under room temperature (RT) or cold exposure (CE) condition. A total of 5636 nuclei were integrated, including 3618 and 2018 nuclei from eight and eight animals for the two conditions, respectively. (**F**) Violin plot of the marker gene expression levels of each mPRAT adipocyte subpopulation under RT or CE condition. (**G**) Differentially expressed genes (DEGs) of each adipocyte subpopulation under RT or CE condition. The top eight genes with the highest fold change (FC) are labeled. (**H–J**) Gene regulatory network (GRN) and functional pathway enrichment comparison between RT and CE conditions within each of the mPRAT-ad1, mPRAT-ad2, and mPRAT-ad3 subpopulations. Bar graphs illustrate the most enriched functional pathways according to the GO, KEGG, Reactome, and WikiPathways databases.

The online version of this article includes the following figure supplement(s) for figure 5:

**Figure supplement 1.** Additional analysis on the medial region of perirenal adipose tissue (mPRAT) under room temperature (RT) and cold exposure (CE) conditions.

in mPRAT (highlighted in gray in *Figure 6C and D*). By label transfer, we found that most of the mPRAT-ad1 adipocytes that did not overlap with iWAT BeAs were more similar to mPRAT-ad2 of the room temperature dataset (highlighted in gray in *Figure 6—figure supplement 2K*), suggesting that they may be derived from mPRAT-ad2 adipocytes that had undergone brown-to-white conversion. Thus, mPRAT BAs that arise from the mPRAT-ad2 subpopulation after cold exposure have a distinct transcriptome to that of cold-induced BeAs in iWAT. Analysis of the hierarchical relationships between different adipocyte subpopulations in the three fat depots following cold exposure revealed that both the BA and WA populations of mPRAT were more closely related to iBAT than iWAT populations (*Figure 6E*). As the ASPC population were rather similar in the different depots at the different temperature conditions (*Figure 5—figure supplement 1H*, *Figure 6—figure supplements 1H* and *2I*), we performed cell–cell communication analysis on the populations of adipocyte in the three depots based on ligand, receptor, and cofactor interactions using the CellChat algorithm (*Jin et al., 2021*). At room temperature, mPRAT adipocyte subpopulations showed greater and stronger intercellular communication than subpopulations in iBAT or iWAT (*Figure 6F and G*), which may be related to the ongoing brown-to-white conversion in mPRAT at room temperature. In contrast, cold exposure reverted these relationships, reducing the number and strength of interactions in mPRAT while increasing interactions in iBAT and iWAT (*Figure 6F and G*). Thus, both under room temperature and cold exposure conditions, mPRAT adipocytes display very different levels of cell–cell communication compared to iBAT and iWAT adipocytes.

## Discussion

Brown-to-white conversion of BAs, also known as 'whitening', has been observed in BAT of a number of mammalian species, including rat (*Florez-Duquet et al., 1998*; *Horan et al., 1988*; *McDonald et al., 1988*), rabbit (*Derry et al., 1972*; *Huang et al., 2022*), bovine and ovine (*Basse et al., 2015*; *Casteilla et al., 1989*; *Gemmell et al., 1972*), and human (*Rogers, 2015*). On the other hand, BA whitening does not occur in mouse iBAT as shown in the present and previous studies (*Huang et al., 2022*). In PRAT, a continuous replacement of BAs by WAs during aging has been suggested based on analysis of adipocyte morphology by a study on the Japanese population (*Tanuma et al., 1976*). Also, so-called 'dormant' BAT has been observed throughout the PRAT depot of an adult kidney donor (*Jespersen et al., 2019*) and habituation to low-temperature environment has been reported to result in the acquisition of BA-like morphology and UCP1 expression in human PRAT (*Efremova et al., 2020*). In the present study, we used snRNA-seq to explore the transcriptome dynamics of mouse PRAT, revealing the existence of brown-to-white conversion in the medial region of the tissue during postnatal development, as confirmed by the loss of UCP1 protein in genetically traced adipocytes. Non-shivering thermogenesis negatively correlates with body mass in a large number of eutherian mammals, as many of them have developed other strategies to cope with low environmental temperatures, such as hibernation and torpor (*Oelkrug et al., 2015*). On the other hand, small-sized mammals with large body surface to volume ratio rely more heavily on BAT for thermogenesis, which may possibly explain why iBAT whitening does not occur in mice. Although our demonstration of a whitening process in the mouse mPRAT would seem to run against this notion, its deep visceral location

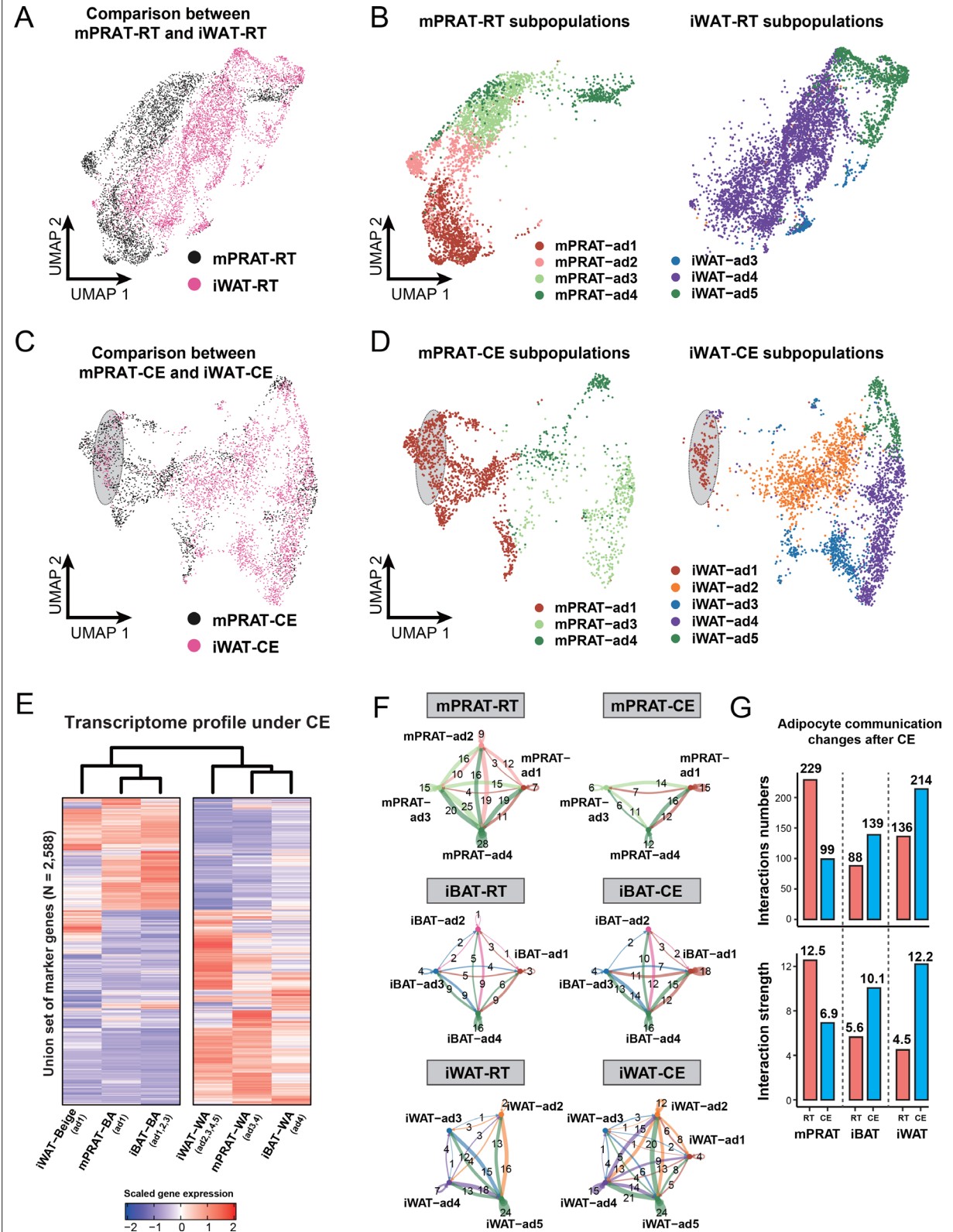

**Figure 6.** mPRAT-ad2 adipocytes have different transcriptomes from inguinal white adipose tissue (iWAT) beige adipocytes (BeAs). Integrated (**A**) and separated (**B**) Uniform Manifold Approximation and Projection (UMAP) of the adipocytes in the medial region of perirenal adipose tissue (mPRAT) and iWAT of 6-month-old C57BL/6J male mice kept under room temperature condition. Integrated (**C**) and separated (**D**) UMAP of the adipocytes in mPRAT and iWAT of 6-month-old C57BL/6J male mice kept under cold exposure condition. The gray-shaded oval was determined by the distribution

*Figure 6 continued on next page*

*Figure 6 continued*

of iWAT-ad1 population in (**D**). (**E**) Heat map illustrating the transcriptome of the brown adipocytes (BAs), BeAs, and white adipocytes (WAs) of mPRAT, interscapular brown adipose tissue (iBAT), and iWAT under cold exposure condition. Overall similarity comparison was illustrated by hierarchical clustering. (**F, G**) Cell–cell communication analysis by CellChat in each adipocyte population of mPRAT, iBAT, and iWAT under room temperature or cold exposure condition. The interaction number is labeled on the corresponding linked lines in (**F**). The communication numbers and strength are shown as bar graphs in (**G**).

The online version of this article includes the following figure supplement(s) for figure 6:

**Figure supplement 1.** Single-nucleus RNA sequencing (snRNA-seq) data analysis of interscapular brown adipose tissue (iBAT) under room temperature (RT) and cold exposure (CE) conditions.

**Figure supplement 2.** Single-nucleus RNA sequencing (snRNA-seq) data analysis of inguinal white adipose tissue (iWAT) under room temperature and cold exposure conditions.

suggests that the thermogenic capacity of this tissue may have other physiological functions. The fact that the entire $Ucp1^-$&$Cidea^+$ mPRAT-ad2 subpopulation was able to become $Ucp1^+$ under short-term cold exposure suggests that a significant portion of mPRAT adipocytes retain thermogenic capacity.

Analysis of the transcriptomes of the different adipocyte subpopulations in mPRAT revealed that their ability to acquire a thermogenic phenotype in response to cold exposure was negatively correlated with the extent to which they expressed WAT marker genes. Thus, while *Cyp2e1*, *Slit3*, *Aldh1a1*, and *Lep* were all expressed by the majority of cold-resistant adipocytes in mPRAT-ad4, only some of the highly cold-responsive $Cidea^+$ cells in mPRAT-ad2 expressed *Cyp2e1* and *Slit3* and none expressed *Aldh1a1* or *Lep*. On the other hand, $Cidea^-$ cells in mPRAT-ad3, a small fraction of which can responsd to cold, were mostly positive for *Cyp2e1* and *Slit3*, but did not express *Aldh1a1* or *Lep*. The fact that mPRAT-ad2 cells express markers of both BA and WA, together with an absence of unique markers for this subpopulation, is in line with their intermediate and plastic phenotype. Our tracing studies showed that only a rather small fraction of adipocytes undergoing brown-to-white conversion in postnatal life expressed mPRAT-ad3 and -ad4 markers CYP2E1 and ALDH1A1, suggesting that the majority of mPRAT-ad3 and -ad4 WAs in PRAT must derive from the differentiation of ASPCs. This also agrees with them being most similar to iBAT-ad4 cells that also derive from ASPCs, given the lack of brown-to-white conversion in the iBAT depot. ASPCs in both PRAT and iBAT were also very similar. We cannot at present exclude the possibility that the limited cold responsiveness of mPRAT-ad3 could be due to inappropriate cold exposure for the induction of a robust thermogenic phenotype. We also note that, although *Cyp2e1* and *Aldh1a1* are always co-expressed in iBAT adipocytes (*Sun et al., 2020*), only *Aldh1a1*, not *Cyp2e1*, distinguished cold-responsive from unresponsive adipocytes in mPRAT, indicating distinct expression patterns and possibly functions for these genes in the two fat depots.

mPRAT-ad2 cells shared with iWAT BeAs the capacity to upregulate *Ucp1* expression and acquire a thermogenic phenotype in response to cold (*Petrovic et al., 2010*; *Wu et al., 2012*). However, none of the subpopulations of iWAT adipocytes showed any similarity to mPRAT-ad2 cells at the transcriptional level. This convergent phenotype may be explained by the different origins of the two depots. Thus, while all adipocyte subpopulations in iWAT arise by differentiation from ASPCs, mPRAT-ad2 adipocytes are derived from brown-to-white conversion of BAs and thus retain a number of molecular signatures from their parental cells. Even after cold exposure, the transcriptome of thermogenic BAs in PRAT (mPRAT-ad1) are more similar to BAs in iBAT than iWAT. These BAs are likely composed of cells from three different origins: recruited mPRAT-ad2 adipocytes, pre-existing mPRAT-ad1 cells similar to iBAT BAs, and BAs that arise from cold-induced de novo adipogenesis. iWAT BeAs derived by cold induction are mainly derived from *Sma*-expressing precursor cells (*Berry et al., 2016*), although interconversion between BeAs and WAs has also been observed after cold exposure (*Rosenwald et al., 2013*). Distinct cellular and/or developmental origins may therefore explain the differences observed among cold-induced adipocytes in the three depots. Adipocytes in BAT and iWAT are known to have different developmental origins. While classical BAs from BAT depots are derived from a subpopulation of dermomyotomes that expresses *Myf5*, *En1*, and *Pax7* (*Atit et al., 2006*; *Lepper and Fan, 2010*; *Sanchez-Gurmaches et al., 2012*; *Seale et al., 2008*), iWAT BeAs emerge from multiple origins, including a *Myf5*-negative cell lineage derived from $Myh11^+$ smooth muscle-like precursors (*Long et al., 2014*) and progenitors expressing *Pax3* and/or *Myf5* (*Sanchez-Gurmaches and Guertin, 2014*). On the other hand, BAs in PRAT arise from a lineage of $Pax3^+$ but $Myf5^-$ cells

(*Sanchez-Gurmaches and Guertin, 2014*). Discrepancies in embryonic origin between mPRAT and iWAT beige adipocytes may have established unique transcriptome signatures that persist during postnatal life. Another possibility is that the kinetics of thermogenic induction, a.k.a. browning, may be different between the two depots such that longer cold exposure results in more similar transcriptional profiles in the two tissues.

One possible limitation of our study is that we only used mice from one genetic background (C57BL/6J), which is known to display a metabolic phenotype that is more prone to develop obesity (*Bachmann et al., 2022*). The transcriptomes and trajectories of cells from fat depots of other strains may deviate from our findings. In addition, the oldest mice that we have analyzed in the present study were 6 months old, which is only equivalent to the human adult stage. Future studies should focus on older mice to investigate whether a more complete brown-to-white conversion takes place during aging as well as the effect of cold exposure on this process. Moreover, lack of specific marker genes for the mPRAT-ad2 subpopulation makes it difficult to generate Cre driver strains that can be used to specifically trace the fate of these adipocytes and visualize the whitening process in a more direct way.

In summary, our work identifies a cold-recruitable adipocyte subpopulation in mPRAT derived by brown-to-white conversion of BAs with a transcriptomic profile distinct from iWAT BeAs. The discrepancy observed in this process between pvPRAT and puPRAT could be functionally related to the idiosyncrasies of kidney blood supply and urine transport, respectively. Previous studies have demonstrated the clinical relevance of PRAT in clear cell renal cell carcinoma invasion (*Wei et al., 2021*) and pathological hypertension (*Li et al., 2022*). Future studies should focus on exploring the physiological role of brown-to-white conversion in different regions of mPRAT and the molecular mechanisms that govern its regulation. Understanding why mPRAT has pre-existing BAs that gradually convert to WAs during postnatal development may contribute to the development of novel therapeutic strategies for renal disorders. More generally, harnessing our understanding of brown-to-white adipocyte conversion may lead to strategies to prevent or delay its course and thus enhance the thermogenic capacity of adipose tissues to mitigate the consequences of metabolic disorders, such as obesity and diabetes.

## Materials and methods

### Animals

All animal experiments were performed in compliance with the protocol approved by the Institutional Animal Care and Use Committee (IACUC) of Peking University (Psych-XieM-2) and the Chinese Institute of Brain Research (CIBR-IACUC-035). C57BL/6J mice were housed under a 12:12 hr light/dark cycle with free access to food and water. For all cold exposure experiments, animals were housed individually in a temperature- and light-controlled incubator (Darth Carter, Hefei, China). *Ucp1Cre* (JAX:024670) (*Kong et al., 2014*), *PdfraCreERT2* (JAX:018280), *PdfraCre* (JAX:013148), and Ai14 (JAX:007914) (*Madisen et al., 2010*) strains were purchased from Jackson Laboratory. *Ucp1CreERT2* strain (*Rosenwald et al., 2013*) was a gift from Dr. Zhinan Yin, Jinan University, China. For genetic tracing, tamoxifen (Sigma-Aldrich, Cat# T5648) dissolved in corn oil (Solarbio, Cat# C7030) was intraperitoneally injected at a dose of 2 mg per animal per day for 4 (*UCP1CreERT2*) or 5 (*PdfraCreERT2*) consecutive days. All animals used in the present study are male.

### Dissection of mouse PRAT

lPRAT was separated from the mPRAT by a thin layer of fascia membrane at both the anterior and posterior renal capsule pole. The adrenal gland and surrounding adipose tissue located at the anterior renal capsule pole were surgically removed by cutting along the fascia membrane on the lateral side and the anterior renal hilum edge on the medial side. mPRAT was defined as the adipose tissue located between the anterior renal hilum edge and the posterior renal capsule pole. Within the mPRAT, pvPRAT was defined as the adipose tissue located between the anterior and posterior renal hilum edge that wraps the renal vessels, and puPRAT was defined as the adipose tissue located between the posterior renal hilum edge and the posterior renal capsule pole that wraps the ureter. For histological analysis, the entire PRAT, together with the kidney, was taken for sectioning to accurately define different regions of the tissue using the kidney as a landmark.

## Nuclei isolation for snRNA-seq

Nuclei isolation from adipose tissue was performed according to a published protocol with slight modification (*Van Hauwaert et al., 2021*). Briefly, adipose tissues were minced in a Petri dish containing 500 mL nuclei isolation buffer (NIB) that contains 250 mM sucrose (Solarbio, Cat# S8271), 10 mM HEPES (Solarbio, Cat# H8090), 1.5 mM MgCl$_2$ (Sigma-Aldrich, Cat# M8266), 10 mM KCl (Sigma-Aldrich, Cat# P5405), 0.001% IGEPAL CA-630 (NP-40) (Sigma-Aldrich, Cat# I3021), 0.2 mM DTT (Sigma-Aldrich, Cat# D9779), and 1 U/µL RNase inhibitor (Solarbio, Cat# R8061) in DEPC-treated water. The samples were further homogenized using a 2 mL Dounce homogenizer (Sigma-Aldrich, Cat# D8938), applying three strokes with the loose pestle. The homogenate was filtered through a 70 µm cell strainer (Falcon, Cat# 352350) and centrifuged at 500 × *g* for 5 min at 4°C. The nuclei pellet was resuspended in 2 mL NIB and centrifuged at 300 × *g* for 3 min at 4°C. Finally, the nuclear pellet was resuspended in 300 µL nuclei resuspension buffer that contains 2% BSA (Sigma-Aldrich, Cat# A1933), 1.5 mM MgCl$_2$, and 1 U/µL RNase inhibitor in PBS. All solutions were sterile filtered prior to use. All procedures were performed on ice.

## snRNA-seq library construction and sequencing

Nuclei suspensions were stained with 10 µg/mL 7-AAD (Invitrogen, Cat# A1310) and sorted on a BD Fusion Flow Cytometer (BD FACSMelody4-Way Cell Sorter) with a 100 µm nozzle. Nuclei were counted using a Countstar Automated Cell Counter (Countstar Rigel S5) and approximately 20,000 nuclei per sample were loaded into the 10x Chromium system using the Single-Cell 3′ Reagent Kit v3.1 according to the manufacturer's instructions (10x Genomics). GEM-Reverse Transcription was performed with a thermal cycler using the following program: 53°C for 45 min, 85°C for 5 min, and held at 4°C. After reverse transcription and cell barcoding, emulsions were lysed and cDNA was isolated and purified with Cleanup Mix containing DynaBeads and SPRIselect reagent (Thermo Scientific), followed by PCR amplification. The cDNA was then evaluated using an Agilent Bioanalyzer. Enzymatic fragmentation and size selection were applied to optimize the cDNA amplicon size. Then, the P5, P7, i7, and i5 sample indexes and TruSeq Read 2 (read 2 primer sequence) were added by end repair, A-tailing, adaptor ligation, and PCR. The final libraries containing the P5 and P7 primers were sequenced by an Illumina NovaSeq 6000 sequencer with 150 bp paired-end reads, aiming for a coverage of approximately 50,000 raw reads per nucleus.

## snRNA-seq data analysis

Raw fastq sequencing data were processed using Cell Ranger version 7.0.0 with the '*--include-introns*' parameter to include reads mapped to introns from Ensembl annotation. The reference genome used for mouse was mm10. The raw nuclei-gene count matrices were processed using CellBender version 0.2.2 (*Fleming et al., 2023*) to remove ambient RNA contamination and empty droplets. The processed count matrices that include validated nuclei were used for downstream analyses with Seurat version 4.3.0 (*Hao et al., 2021*).

For single-dataset preprocessing, we removed low-quality nuclei according to the following criteria: (1) nuclei contain less than 500 or more than 6000 genes; (2) nuclei contain more than 15% of total reads from mitochondrial genes; and (3) nuclei contain less than 1000 or more than 50,000 UMI counts. Then, genes expressed in less than 10 nuclei were removed. Doublet score was calculated using DoubletFinder version 2.0.3 (*McGinnis et al., 2019*), and 8% of the nuclei that had the highest doublet score were removed according to the recommendation of 10x Genomics. The '*LogNormalize*' function was used to perform normalization, and z-scale was performed using the '*ScaleData*' function. Highly variable genes (HVGs) were calculated using the Variance-Stabilizing Transformation algorithm, and the top 2000 HVGs for each dataset were used for downstream analysis. Kidney-related cells and lymphocytes that come from contaminations during dissection were removed from the datasets prior to further analysis.

For multiple-dataset integration, the Canonical Correlation Analysis algorithm was used. Specifically, the '*SelectIntegrationFeatures*' and '*FindIntegrationAnchors*' functions were used to find integration features based on HVGs, and multiple datasets were integrated using the '*IntegrateData*' function in Seurat. Principal component analysis (PCA) was performed using the '*RunPCA*' function to obtain the first 30 principal components. Nuclei were then projected into the low-dimensional space using the Uniform Manifold Approximation and Projection (UMAP) algorithm. Nuclei clusters

were defined using the Louvain algorithm via the '*FindClusters*' function. Data embedding for UMAP clustering was performed according to the top 2000 variable genes and the first 30 principal components. Marker genes for each cluster were calculated using the '*FindAllMarkers*" function by a Wilcox test (log2 fold change > 0.25) and clusters were manually annotated either based on marker genes reported in literature or by searching top markers genes against the PanglaoDB database (*Franzén et al., 2019*). Marker gene lists for all datasets are summarized in *Supplementary file 1* (https:// github.com/HouyuZhang/PRAT_project/blob/main/Supplementary_Table1_Marker_gene_lists_for_ all_datasets.xlsx; *Zhang, 2024b*).

The low-dimensional dataset from Seurat was applied to Monocle version 3 (*Cao et al., 2019*) for trajectory inference using the partitioned approximate graph abstraction algorithm. A principal graph-embedding procedure based on the SimplePPT algorithm was applied via the '*learn_graph*' function to learn a principal graph representing the possible paths that the nuclei can take as they progress. Once the trajectories were defined, we manually selected the root node that resides specific population. For the time-series mPRAT trajectory analysis, we defined root node resides the *Ucp1* highest as the starting point. Pseudo-time for each nucleus was computed as its geodesic distance back to our defined root nodes in the trajectory using the principal graph as a guide. Then, spatial correlation analysis by the Moran's I test was conducted via the '*graph_test*' function to find DEGs across our defined trajectory.

For cluster similarity analysis, top 50 marker genes ranked by q-value from the reference cluster were selected as a gene module, then the module score for each cell of the query cluster was calculated using the '*AddModuleScore_UCell*' function (*Andreatta and Carmona, 2021*). A label transfer method from Seurat was used to learn annotations among snRNA-seq dataset. Specifically, the '*FindTransferAnchors*' function was used between reference dataset and query dataset to find anchors, which were further used to annotate each cell in query dataset by the annotation in reference dataset by the '*TransferData*' function.

DEGs between two clusters were calculated using the '*FindMarkers*' function in Seurat. Specifically, genes expressed in more than 10% of cells of either cluster were compared by a Wilcoxon rank-sum test. The '*compareCluster*' and '*enrichGO*' functions of the clusterProfiler version 4.8.1 (*Wu et al., 2021*) were used to find enriched GO pathways using the species-specific annotation database ( org.Mm.eg.db). The Benjamini–Hochberg methods were used for p-value correction with 0.05 as a threshold. For mPRAT-RT and mPRAT-CE analysis, the Metascape website version 3.5 (*Zhou et al., 2019*) was used to predict GRN and functional enrichment against GO, KEGG, Reactome, and WikiPathways database.

The cell–cell communication analysis was conducted with CellChat version 1.6.0 using the mouse database. The '*computeCommunProbPathway*' function was used to compute the communication probability on signaling pathway level by summarizing all related ligands or receptors, and p-value<0.05 was used to determine significant interactions.

## Gene expression analysis

Total RNA was extracted from snap-frozen adipose tissues using the TRIzol reagent (Invitrogen, Cat# 15596018). cDNA was synthesized from 1 μg of RNA using the H Minus cDNA first-strand synthesis kit (Thermo Scientific, Cat# K1652). Relative mRNA expression was determined by quantitative PCR using the PowerUp SYBR TM Green Master Mix (Applied Biosystems, Cat# A25742) in a CFX96TM Real-Time PCR Detection System (Bio-Rad). Cq values were normalized to levels of *Ywhaz* using the ΔΔ-Ct method. The primer sequences are as follows: mouse *Ucp1* forward 5'-GGCCTCTACGACTCAG TCCA-3', reverse 5'-TAAGCCGGCTGAGATCTTGT-3'; mouse *Ywhaz* forward 5'-CAGTAGATGGAG AAAGATTTGC-3', reverse 5'-GGGACAATTAGGGAAGTAAGT-3'.

## Immunoblotting and antibodies

0.3 mL of 100% ethanol was added to the collected interphase and lower organic layer from the TRIzol lysate and incubated for 3 min. The mixture was centrifuged at 2000 × *g* for 5 min at 4°C to pellet the DNA. The phenol–ethanol supernatant was transferred to a new tube and 1.5 mL of isopropanol was added. After a 10 min incubation, the mixture was centrifuged at 12,000 × *g* for 10 min at 4°C to pellet the proteins. The pellet was washed three times with a washing solution consisting of 0.3 M guanidine hydrochloride (Solarbio, Cat# G8070) in 95% ethanol, followed by one more wash with 100% ethanol.

The protein pellet was air-dried and resuspended in 200 µL of 1% SDS (Solarbio, Cat# S8010) at 50°C in a heating block. Protein concentration was determined using a BCA protein assay kit (Solarbio, Cat# PC0020) in a Cytation 5 imaging reader (Thermo Fisher). Protein samples were prepared using the Laemmli buffer (Bio-Rad, Cat# 1610747), supplemented with 20% 2-mercaptoethanol, and boiled at 98°C. Reducing SDS-PAGE gel was made using the SDS-PAGE Gel kit (Bio-Rad, Cat# 1610183). 15 µg sample was loaded into SDS-PAGE gel and transferred onto PVDF membranes (Bio-Rad, Cat# 1620177). The blots were processed with a chemiluminescence HRP Substrate (Millipore, Cat# WBKLS05000) and imaged with the Amersham ImageQuant 800 western blot imaging system. The intensity of target bands was quantified using ImageJ version 1.53k.

The antibodies and the working concentrations are as follows: UCP1: 1:1000 dilution (Sigma-Aldrich, Cat# U6382), α-tubulin: 1:1000 dilution (Santa Cruz, Cat# sc-8035), and anti-rabbit IgG: 1:2000 dilution (Cell Signaling, Cat# 7074S).

## Hematoxylin and eosin (HE) and Tunel staining

Freshly collected adipose tissues were fixed in 4% paraformaldehyde overnight at 4°C and rinsed with PBS. After serial dehydration in ethanol, tissues were embedded in paraffin and sectioned into 5-µm-thick sections for staining with the HE solution (Servicebio, Cat# G1003). For Tunel staining, sections were deparaffinized and rehydrated by serial xylene and ethanol before antigen retrieval. Then, sections were permeabilized using PBS with 0.1% Triton and blocked with 3% BSA. Tunel Assay Kit (Servicebio, Cat# G1501) was used to detect DNA breaks formed during apoptosis. Perilipin-1 at 1:100 dilution (Cell Signaling, Cat# 9349S) was used to stain all adipocytes. DAPI was used to stain nuclei.

## Immunofluorescence staining

Whole-mount immunofluorescence was performed as previously described (*Jiang et al., 2018*). Briefly, mice were anesthetized and perfused with PBS, before being sacrificed for adipose tissue dissection followed by overnight fixation in 1% paraformaldehyde at 4°C. The tissues were washed in PBS containing 0.01% Tween-20 (PBST) before being embedded in 5% low-melting agarose and sectioned into 100–400-µm-thick slices using a Vibratome (Leica, VT1200). Slices were blocked and permeabilized using PBST supplemented with 5% donkey serum and 0.3% Triton X at room temperature for 30 min. If necessary, the M.O.M. Blocking Reagent (Vector, Cat# MKB-2213-1) was applied to block mouse endogenous IgG for background reduction. The following antibodies and chemicals were used: rabbit-host UCP1, 1:100 (Sigma-Aldrich, Cat# U6382), mouse-host UCP1, 1:100 (Invitrogen, Cat# MA5-31534), mouse-host ALDH1A, 1:50 (Proteintech, Cat# 60171-1-Ig), rabbit-host CYP2E1, 1:50 (Abcam, Cat# ab28146), rabbit-host PLIN1, 1:100 (Cell Signaling, Cat# 9349), Alexa Fluor 405 mouse IgG H&L, 1:1000 (Invitrogen, Cat# A48257), Alexa Fluor 647 rabbit IgG H&L, 1:1000 (Invitrogen, Cat# A31573), BODIPY 493/503, 1:500 (Invitrogen, Cat# D3922), and DAPI, 1:20 (Solarbio, Cat# C0065). Slices were imaged using an Airyscan 2 LSM 900 confocal microscope (ZEISS). Images were processed and analyzed using the Imaris software (Bitplane version 9.0.1).

## Whole-tissue immunostaining and optical clearing

Whole-tissue optical clearing and immunostaining were performed following the previously published ImmuView method (*Ding et al., 2019*). Briefly, mice were anesthetized and perfused with PBS. The kidney and all surrounding adipose tissues (mPRAT and lPRAT) were collected and fixed in PBS containing 0.5% PFA and 10% sucrose at room temperature for 2 hr. After serial dehydration and rehydration in methanol of different concentrations, the tissues were permeabilized and blocked overnight using PBS containing 0.2% Triton X-100, 10% DMSO (Sigma-Aldrich, Cat# D8418), 5% donkey serum, and 10 mM EDTA-Na at room temperature, followed by immunolabeling with UCP1 primary antibody (1:100, Sigma-Aldrich, Cat# U6382) in a PBS-based dilution buffer containing 0.2% Tween-20, 10 mg/mL heparin (Solarbio, Cat# H8060), 5% donkey serum, and 10 mM EDTA-Na (pH 8.0) for 3 d at room temperature. Then, the tissues were washed with a washing buffer (PBS with 0.2% Tween-20, 10 mg/mL heparin, and 10 mM EDTA-Na) for 12 hr at 37°C. Washing buffer was changed every 2 hr during the washing period. Alexa Fluor 555 rabbit IgG H&L secondary antibody (1:1000, Invitrogen, Cat# A31572) was used for primary antibody detection by incubation at room temperature for 3 d, followed by a 2-day washing at 37°C with washing buffer being changed every 8 hr. For

tissue clearing, the immunolabeled tissues were embedded in 0.8% agarose and incubated at room temperature with increasing concentrations of methanol. The tissue blocks were then incubated at room temperature with a mixture of dichloromethane and methanol (2:1) for 4 hr, followed by incubation in 100% dichloromethane for 1.5 hr. The tissue blocks were finally incubated at room temperature with 100% dibenzyl ether (DBE) for 24 hr. All incubation steps were performed with gentle shaking.

Samples were imaged with an Ultramicroscope II light-sheet microscope (LaVision Biotec) equipped with a sCMOs camera (Andor Neo). Images were acquired with the ImspectorPro software (LaVision BioTec). Samples were placed in an imaging reservoir filled with DBE and illuminated from the side by the laser light sheet. The samples were scanned with the 640 nm laser, with a step size of 3 μm for ×4 objectives. Images were processed using the Imaris software (Bitplane version 9.0.1).

### Statistical analysis

Statistical analyses were performed using GraphPad Prism version 8.0.0. Results were presented as mean ± standard deviation (SD). One-way or two-way ANOVA was used to test statistical significance for qPCR, western blot, and immunofluorescence quantification as specified in the corresponding figure legends. The level of statistical significance was assigned as $*p<0.05$, $**p<0.01$, $***p<0.001$, $****p<0.0001$.

## Acknowledgements

The authors thank Dr. Zhinan Yin, Jinan University, for providing the Ucp1CreERT2 mouse strain. The authors also thank Hong Jin and Dr. Jing Yang, Peking University, for technical support on nuclei extraction and the ImmuView method, respectively. This work was supported by research grants to CFI from Peking University, Chinese Institute for Brain Research, Beijing, Swedish Cancer Society (Cancerfonden, contract no. 222135Pj01H) and Swedish Research Council (Vetenskapsrådet, contract no. 2020-01923_3); and a startup grant to MX from Swedish Research Council (Vetenskapsrådet, contract no. 2021-01805).

## Additional information

### Funding

| Funder | Grant reference number | Author |
| --- | --- | --- |
| Chinese Institute for Brain Research | | Carlos F Ibáñez |
| Peking University | | Carlos F Ibáñez |
| Cancerfonden | 222135Pj01H | Carlos F Ibáñez |
| Vetenskapsrådet | 2020-01923_3 | Carlos F Ibáñez |
| Vetenskapsrådet | 2021-01805 | Meng Xie |

The funders had no role in study design, data collection and interpretation, or the decision to submit the work for publication.

### Author contributions

Houyu Zhang, Data curation, Formal analysis, Investigation, Methodology; Yan Li, Data curation; Carlos F Ibáñez, Conceptualization, Supervision, Funding acquisition, Writing – review and editing; Meng Xie, Conceptualization, Supervision, Funding acquisition, Writing - original draft, Writing – review and editing

### Author ORCIDs

Meng Xie http://orcid.org/0000-0002-6388-9789

## Ethics

All animal experiments were performed in compliance with the protocol approved by the Institutional Animal Care and Use Committee (IACUC) of Peking University (Psych-XieM-2) and the Chinese Institute of Brain Research (CIBR-IACUC-035).

Reviewer #1 (Public Review): https://doi.org/10.7554/eLife.93151.3.sa1
Reviewer #2 (Public Review): https://doi.org/10.7554/eLife.93151.3.sa2
Author Response https://doi.org/10.7554/eLife.93151.3.sa3

---

# Additional files

## Supplementary files

• MDAR checklist

• Supplementary file 1. Marker gene lists for all datasets.

## Data availability

All raw sequencing data reported in the present study were deposited in the Gene Expression Omnibus (GEO) database under accession number GSE241800. All codes to repeat snRNA-seq analysis were in the GitHub repository (https://github.com/HouyuZhang/PRAT_project, copy archived at *Zhang, 2024a*).

The following dataset was generated:

| Author(s) | Year | Dataset title | Dataset URL | Database and Identifier |
|---|---|---|---|---|
| Zhang H, Li Y, Ibáñez CF, Xie M | 2023 | Transcriptome-wide profiling of different mouse adipose tissues during aging using snRNA-seq | https://www.ncbi.nlm.nih.gov/geo/query/acc.cgi?acc=GSE241800 | NCBI Gene Expression Omnibus, GSE241800 |

The following previously published dataset was used:

| Author(s) | Year | Dataset title | Dataset URL | Database and Identifier |
|---|---|---|---|---|
| Sárvári AK, Van Hauwaert EL, Markussen LK, Gammelmark E, Marcher A, Ebbesen MF, Nielsen R, Brewer JR, Madsen JG, Mandrup S | 2021 | Plasticity of epididymal adipose tissue in response diet-induced obesity at single-nucleus resolution | https://www.ncbi.nlm.nih.gov/geo/query/acc.cgi?acc=GSE160729 | NCBI Gene Expression Omnibus, GSE160729 |

---

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
