## [Editor Report · eLife assessment]

This study presents a **valuable** finding on the process of brown-to-white adipogenic transdifferentiation within the perirenal adipose depot. The evidence supporting the claims is **convincing**, although limited sequencing depth of single nuclei and lack of regulatory insights somewhat lessens the impact of these findings. The work will be of interest to adipose tissue biologists.

---

## [Referee Report · Reviewer #1 (Public Review)]

Summary: In this manuscript, the authors performed single nucleus RNA-seq for perirenal adipose tissue (PRAT) at different ages. They concluded a distinct subpopulation of adipocytes arises through beige-to-white conversion and can convert to a thermogenic phenotype upon cold exposure.

Strengths: PRAT adipose tissue has been reported as an adipose tissue that undergoes browning. This study confirms that beige-to-white and white-to-beige conversions also exist in PRAT, as previously reported in the subcutaneous adipose tissue.

Weaknesses:

(1) There is overall a disconnection between single nucleus RNA-seq data and the lineage chasing data. No specific markers of this population have been validated by staining.

(2) It would be nice to provide more evidence to support the conclusion shown in lines 243 to 245 "These results indicated that new BAs induced by cold exposure were mainly derived from UCP1- adipocytes rather than de novo ASPC differentiation in puPRAT". Pdgfra-negative progenitor cells may also contribute to these new beige adipocytes.

(3) The UCP1Cre-ERT2; Ai14 system should be validated by showing Tomato and UCP1 co-staining right after the Tamoxifen treatment.

---

## [Referee Report · Reviewer #2 (Public Review)]

Summary:

In the present manuscript, Zhang et al utilize single-nuclei RNA-Seq to investigate the heterogeneity of perirenal adipose tissue. The perirenal depot is interesting because it contains both brown and white adipocytes, a subset of which undergo functional "whitening" during early development. While adipocyte thermogenic transdifferentiation has been previously reported, there remains many unanswered questions regarding this phenomenon and the mechanisms by which it is regulated.

Strengths:

The combination of UCP1-lineage tracing with the single nuclei analysis allowed the authors to identify four populations of adipocytes with differing thermogenic potential, including an "whitened" adipocyte (mPRAT-ad2) that retains the capacity to rapidly revert to a brown phenotype upon cold exposure. They also identify two populations of white adipocytes that do not undergo browning with acute cold exposure.

Anatomically distinct adipose depots display interesting functional differences, and this work contributes to our understanding of one of the few brown depots present in humans.

Weaknesses:

The most interesting aspect of this work is the identification of a highly plastic mature adipocyte population with the capacity to switch between a white and brown phenotype. The authors attempt to identify the transcriptional signature of this ad2 subpopulation, however the limited sequencing depth of single nuclei somewhat lessens the impact of these findings. Furthermore, the lack of any form of mechanistic investigation into the regulation of mPRAT whitening limits the utility of this manuscript. However, the combination of well-executed lineage tracing with comprehensive cross-depot single-nuclei presented in this manuscript could still serve as a useful reference for the field.

---

## [Author Response]

The following is the authors’ response to the original reviews.

**eLife assessment**
This study presents a valuable finding on the distinct subpopulation of adipocytes during brown-to-white conversion in perirenal adipose tissue (PRAT) at different ages. The evidence supporting the claims of the authors is convincing, although specific lineage tracing of this subpopulation of cells and mechanistic studies would expand the work. The work will be of interest to scientists working on adipose and kidney biology.
**Public Reviews:**

**Reviewer #1 (Public Review):**
Summary:In this manuscript, the authors performed single nucleus RNA-seq for perirenal adipose tissue (PRAT) at different ages. They concluded a distinct subpopulation of adipocytes arises through brown-to-white conversion and can convert to a thermogenic phenotype upon cold exposure.Strengths:PRAT adipose tissue has been reported as an adipose tissue that undergoes browning. This study confirms that brown-to-white and white-to-beige conversions also exist in PRAT, as previously reported in the subcutaneous adipose tissue.

Response: We thank the reviewer for summarizing the strengths of our manuscript. However, we would like to clarify two points here. First, PRAT has been reported as a visceral adipose depot that contains brown adipocytes and a process of continuous replacement of brown adipocytes by white adipocytes has been previously suggested based on histological assessment. There is no evidence that PRAT undergoes browning, unless cold exposure is involved. Second, unlike the brown-to-white conversion, white-to-beige conversion in PRAT was not observed under normal conditions. The adipocyte population that arises from brown-to-white conversion (mPRAT-ad2) can respond to cold and restore their UCP1 expression. However, the adipocytes that arise from the mPRAT-ad2 subpopulation after cold exposure have a distinct transcriptome to that of cold-induced beige adipocyte in iWAT (Figure S7K) and are more related to iBAT brown adipocytes (Figure 6E). Therefore, it is more of a white-to-brown conversion in PRAT upon cold exposure rather than white-to-beige conversion and the underlying mechanism is likely different from the white-to-beige conversion in the subcutaneous adipose tissue.

Weaknesses:(1) There is overall a disconnection between single nucleus RNA-seq data and the lineage chasing data. No specific markers of this population have been validated by staining.

Response: We are not sure what “this population” refers to. We assume that it is the Ucp1-&Cidea+ mPRAT-ad2 adipocyte subpopulation. If so, we did not identify specific markers for these adipocytes as shown in Figure 1H and statements in the Discussion section. mPRAT-ad2 is negative for Ucp1 and Cyp2e1, which are markers for mPRAT-ad1 and mPRAT-ad3&4, respectively. To visualize the mPRAT-ad2 adipocytes on tissue sections, we collected pvPRAT and puPRAT of Ucp1CreERT2;Ai14 mice one day after tamoxifen injection and stained with CYP2E1 antibody and BODIPY. The Tomato-&CYP2E1-&BODIPY+ cells represent the mPRAT-ad2 adipocytes. Based on such strategy, we revealed a significantly higher percentage of mPRAT-ad2 cells in puPRAT than pvPRAT (presented as Figure S3E in the revised manuscript).

(2) It would be nice to provide more evidence to support the conclusion shown in lines 243 to 245 "These results indicated that new BAs induced by cold exposure were mainly derived from UCP1- adipocytes rather than de novo ASPC differentiation in puPRAT". Pdgfra-negative progenitor cells may also contribute to these new beige adipocytes.

Response: We stained pvPRAT and puPRAT of the PdgfraCre;Ai14 mice with the adipocyte marker Plin1 and observed a 100% overlap between the tdTomato signal and the Plin1 staining, after examining a total of 832 and 628 adipocytes in pvPRAT and puPRAT of two animals (Figure S4). Plin1 stains all adipocytes, while the endogenous tdTomato labels both the adipocytes and blood vessels. This result suggests that all adipocytes in mPRAT are derived from Pdgfra-expressing cells, which is in line with a previous study that integrated several single-cell RNA sequencing data sets and showed that Pdgfra is expressed by virtually all ASPCs (Ferrero et al., 2020).

Also, we would like to point out that the cold-induced adipocytes in mPRAT resemble more to the brown adipocytes of iBAT than the beige adipocytes of iWAT (Figure 6E and S7K).

Ferrero, R., Rainer, P., and Deplancke, B. (2020). Toward a Consensus View of Mammalian Adipocyte Stem and Progenitor Cell Heterogeneity. Trends Cell Biol 30, 937-950.

(3) The UCP1Cre-ERT2; Ai14 system should be validated by showing Tomato and UCP1 co-staining right after the Tamoxifen treatment.

Response: We collected pvPRAT and puPRAT of 1- and 6-month-old Ucp1CreERT2;Ai14 mice one day after the last tamoxifen injection and stained with UCP1 antibody to check the overlap between the Tomato and UCP1signal. All Tomato+ cells were UCP1+, indicating 100% specificity of the Ucp1CreERT2; and the labelling efficiency was over 93% at both time points for both regions (Figure S3C-D).

**Reviewer #2 (Public Review):**
Summary:In the present manuscript, Zhang et al utilize single-nuclei RNA-Seq to investigate the heterogeneity of perirenal adipose tissue. The perirenal depot is interesting because it contains both brown and white adipocytes, a subset of which undergo functional "whitening" during early development. While adipocyte thermogenic transdifferentiation has been previously reported, there remain many unanswered questions regarding this phenomenon and the mechanisms by which it is regulated.Strengths:The combination of UCP1-lineage tracing with the single nuclei analysis allowed the authors to identify four populations of adipocytes with differing thermogenic potential, including a "whitened" adipocyte (mPRAT-ad2) that retains the capacity to rapidly revert to a brown phenotype upon cold exposure. They also identify two populations of white adipocytes that do not undergo browning with acute cold exposure.Anatomically distinct adipose depots display interesting functional differences, and this work contributes to our understanding of one of the few brown depots present in humans.Weaknesses:The most interesting aspect of this work is the identification of a highly plastic mature adipocyte population with the capacity to switch between a white and brown phenotype. The authors attempt to identify the transcriptional signature of this ad2 subpopulation, however, the limited sequencing depth of single nuclei somewhat lessens the impact of these findings. Furthermore, the lack of any form of mechanistic investigation into the regulation of mPRAT whitening limits the utility of this manuscript. However, the combination of well-executed lineage tracing with comprehensive cross-depot single-nuclei presented in this manuscript could still serve as a useful reference for the field.

Response: The sequencing depth of our data is comparable, if not better than previously published snRNA-seq studies on adipose tissue (Burl et al., 2022; Sarvari et al., 2021; Sun et al., 2020). Therefore, the depth of our data has reached the limit of the 3’ sequencing methods. Unfortunately, due to size limitation of the adipocytes, it is challenging to sort them for Smart-seq. We suspect that lack of specific markers for mPRAT-ad2 is partly due to its intermediate and plastic phenotype. Regarding the mechanistic regulation of mPRAT whitening, we believe that it is more suitable to leave such investigations for a separate follow-up and more in-depth study.

Burl, R.B., Rondini, E.A., Wei, H., Pique-Regi, R., and Granneman, J.G. (2022). Deconstructing cold-induced brown adipocyte neogenesis in mice. Elife 11. 10.7554/eLife.80167.

Sarvari, A.K., Van Hauwaert, E.L., Markussen, L.K., Gammelmark, E., Marcher, A.B., Ebbesen, M.F., Nielsen, R., Brewer, J.R., Madsen, J.G.S., and Mandrup, S. (2021). Plasticity of Epididymal Adipose Tissue in Response to Diet-Induced Obesity at Single-Nucleus Resolution. Cell Metab 33, 437-453 e435. 10.1016/j.cmet.2020.12.004.

Sun, W., Dong, H., Balaz, M., Slyper, M., Drokhlyansky, E., Colleluori, G., Giordano, A., Kovanicova, Z., Stefanicka, P., Balazova, L., et al. (2020). snRNA-seq reveals a subpopulation of adipocytes that regulates thermogenesis. Nature 587, 98-102. 10.1038/s41586-020-2856-x.

**Recommendations for the authors:**

**Reviewer #1 (Recommendations For The Authors):**
(1) There is overall a disconnection between single nucleus RNA-seq data and the lineage chasing data. No specific markers of this population have been validated by staining.(2) It would be nice to provide more evidence to support the conclusion shown in lines 243 to 245: "These results indicated that new BAs induced by cold exposure were mainly derived from UCP1- adipocytes rather than de novo ASPC differentiation in puPRAT". Pdgfra-negative progenitor cells may also contribute to these new beige adipocytes.(3) The UCP1Cre-ERT2; Ai14 system should be validated by showing Tomato and UCP1 co-staining right after the Tamoxifen treatment.

Please see above for the responses.

**Reviewer #2 (Recommendations For The Authors):**
Without specific lineage tracing it is not possible to conclude that the mPRAT-ad2 population converted to beige with CE. The authors should change this wording from "likely" to "possible".

Response: We have changed the word “likely” to “possible” in the text. Also, we would like to point out that the cold-induced adipocytes in mPRAT resemble more to the brown adipocytes of iBAT than the beige adipocytes of iWAT (Figure 6E and S7K).

The sentence "precursor cells may be less sensitive to environmental temperature and have a limited contribution to mature adipocyte phenotypes through de novo adipogenesis after cold exposure." and others like it should be changed to indicate the acute timeframe of this experiment. It has been shown that the precursors make a more significant contribution to de novo beige adipogenesis with chronic cold exposure.

Response: We have modified the sentence as follows: “precursor cells may be less sensitive to acute environmental temperature drop and have a limited contribution to mature adipocyte phenotypes through de novo adipogenesis after cold exposure”. As mentioned above, the cold-induced adipocytes in mPRAT resemble more to the brown adipocytes of iBAT and therefore may have a different mechanism to the de novo beige adipogenesis with chronic cold exposure.